# Practical $0.385$-Approximation for Submodular Maximization Subject to a Cardinality Constraint

**Murad Tukan**
DataHeroes Israel
murad@dataheroes.ai

**Loay Mualem**
Department of Computer Science
University of Haifa
Haifa Israel
loaymual@gmail.com

**Moran Feldman**
Department of Computer Science
University of Haifa
Haifa Israel
moranfe@cs.haifa.ac.il

## Abstract

Non-monotone constrained submodular maximization plays a crucial role in various machine learning applications. However, existing algorithms often struggle with a trade-off between approximation guarantees and practical efficiency. The current state-of-the-art is a recent $0.401$-approximation algorithm, but its computational complexity makes it highly impractical. The best practical algorithms for the problem only guarantee $1/e$-approximation. In this work, we present a novel algorithm for submodular maximization subject to a cardinality constraint that combines a guarantee of $0.385$-approximation with a low and practical query complexity of $O(n+k^2)$, where $n$ is the size of the ground set and $k$ is the maximum size of a feasible solution. Furthermore, we evaluate the empirical performance of our algorithm in experiments based on the machine learning applications of Movie Recommendation, Image Summarization, and Revenue Maximization. These experiments demonstrate the efficacy of our approach.

## 1  Introduction

In the last few years, the ability to effectively summarize data has gained importance due to the advent of massive datasets in many fields. Such summarization often consists of selecting a small representative subset from a large corpus of images, text, movies, etc. Without a specific structure, this task can be as challenging as finding a global minimum of a non-convex function. Fortunately, many practical machine learning problems exhibit some structure, making them suitable for optimization techniques (either exact or approximate).

A key structure present in many such problems is submodularity, also known as the principle of diminishing returns. This principle suggests that the incremental value of an element decreases as the set it is added to grows. Submodularity enables the creation of algorithms that can provide near-optimal solutions, making it fundamental in machine learning. It has been successfully applied to various tasks, such as social graph analysis [39], adversarial attacks [26, 36], dictionary learning [13], data summarization [32, 34, 35], interpreting neural networks [14], robotics [45, 41], and many more.

To exemplify the notion of submodularity, consider the following task. Given a large dataset, our goal is to identify a subset that effectively summarizes (or covers) the data, with a good representative

set being one that covers the majority of the data. Note that adding an element $s$ to a set $B$ is less beneficial to this goal than adding it to a subset $A \subset B$ due to the higher likelihood of overlapping coverage. Formally, if $\mathcal{N}$ is the set of elements in the dataset, and we define a function $f \colon 2^{\mathcal{N}} \to \mathbb{R}$ mapping every set of elements to its coverage, then, the above discussion implies that, for every two sets $A \subseteq B \subseteq \mathcal{N}$ and element $s \in \mathcal{N} \setminus B$, it must hold that $f(s \mid A) \geq f(s \mid B)$, where $f(s \mid A) \triangleq f(\{s\} \cup A) - f(A)$ denotes the marginal gain of the element $s$ with respect to the set $A$. We say that a set function is *submodular* if it obeys this property.

Unfortunately, maximizing submodular functions is NP-hard even without a constraint [16], and therefore, works on maximization of such functions aim for approximations. Many of these works make the extra assumption that the submodular function $f \colon 2^{\mathcal{N}} \to \mathbb{R}$ is *monotone*, i.e., that for every two sets $A \subseteq B \subseteq \mathcal{N}$, it holds that $f(B) \geq f(A)$. Two of the first works of this kind, by Nemhauser and Wolsey [37] and Nemhauser et al. [38], showed that a greedy algorithm achieves a tight $1 - 1/e$ approximation for the problem of maximizing a non-negative monotone submodular function subject to a cardinality constraint using $O(nk)$ function evaluations, where $n$ is the size of the ground set $\mathcal{N}$ and $k$ is the maximum cardinality allowed for the output set. An important line of work aimed to improve the time complexity of the last algorithm, culminating with deterministic and randomized algorithms that have managed to reduce the time complexity to linear at the cost of an approximation guarantee that is worse only by a factor of $1 - \varepsilon$ [6, 31, 27, 22, 20].

Unfortunately, the submodular functions that arise in machine learning applications are often non-monotone, either because they are naturally non-monotone, or because a diversity-promoting non-monotone regularizer is added to them. Maximizing a non-monotone submodular function is challenging. The only tight approximation known for such functions is for the case of unconstrained maximization, which enjoys a tight approximation ratio of $1/2$ [16, 7]. A slightly more involved case is the problem of maximizing a non-negative (not necessarily monotone) submodular function subject to a cardinality constraint. This problem has been studied extensively. First, Lee et al. [25] suggested an algorithm guaranteeing $(1/4 - \varepsilon)$-approximation for it. This approximation ratio was improved in a long series of works [5, 11, 15, 43], leading to a very recent $0.401$-approximation algorithm due to Buchbinder and Feldman [4], which improved over a previous $0.385$-approximation algorithm due to Buchbinder and Feldman [3]. On the inapproximability side, it has been shown that no algorithm can guarantee a better approximation ratio than $0.478$ in polynomial time [40].

Most of the results in the above-mentioned line of work are only of theoretical interest due to a very high time complexity. The two exceptions are the Random Greedy algorithm of Buchbinder et al. [5] that guarantees $1/e$-approximation using $O(nk)$ queries to the objective function, and the Sample Greedy algorithm of Buchbinder et al. [6] that reduces the query complexity to $O_{\varepsilon}(n)$ at the cost of a slightly worse approximation ratio of $1/e - \varepsilon$.

## 1.1 Our contribution

In this work, we introduce a novel combinatorial algorithm for maximizing a non-negative submodular function subject to a cardinality constraint. Our suggested method combines a practical query complexity of $O(n + k^2)$ with an approximation guarantee of $0.385$, which improves over the $1/e$-approximation of the state-of-the-art practical algorithm. To emphasize the effectiveness of our suggested method, we empirically evaluate it on 3 applications: (i) Movie Recommendation, (ii) Image Summarization, and (iii) Revenue Maximization. Our experiments on these applications demonstrate that our algorithm (Algorithm 3) outperforms the current practical state-of-the-art algorithms.

**Remark.** An independent work that recently appeared on arXiv [12] suggests another $0.385$-approximation algorithm for our problem using $O(nk)$ oracle queries. Interestingly, their algorithm is very similar to a basic version of our algorithm presented in Appendix A. In this work, our main goal is to find ways to speed up this basic algorithm, which leads to our main result. In contrast, the main goal of [12] is to derandomize the basic algorithm and extend it to other constraints.

## 1.2 Additional notation

Let us define some additional notation used throughout the paper. Given an element $u \in \mathcal{N}$ and a set $S \subseteq \mathcal{N}$, we use $S + u$ and $S - u$ as shorthands for $S \cup \{u\}$ and $S \setminus \{u\}$, respectively. Given also a set function $f \colon 2^{\mathcal{N}} \to \mathbb{R}$, we recall that $f(u \mid S)$ is used to denote the marginal contribution of $u$ to

$S$. Similarly, given an additional set $T \subseteq \mathcal{N}$, we define $f(T \mid S) \triangleq f(S \cup T) - f(S)$. Finally, we denote by $\mathbb{OPT}$ an arbitrary optimal solution for the problem we consider.

## 2 Method

In this section, we present our algorithm for non-monotone submodular maximization under cardinality constraints, which is the algorithm used to prove the main theoretical result of our work (Theorem 2.3). We begin with a brief overview of our algorithm. Motivated by the ideas underlying the impractical $0.385$-approximation algorithm of [3], our algorithm comprises three steps:

1. **Initial Solution:** We start by searching for a good initial solution that guarantees a constant approximation to the optimal set. This is accomplished by running the recent deterministic $1/4$-approximation algorithm of Balkanski et al. [2].[1]

2. **Accelerated Local Search (Algorithm 1):** Next, the algorithm aims to find an (approximate) local optimum set $Z$ using a local search method. This can be done using a classical local search algorithm at the cost of $O_\varepsilon(nk^2)$ queries (see Appendix A for more detail). As an alternative, we introduce, in Subsection 2.1, our accelerated local search algorithm FAST-LOCAL-SEARCH (Algorithm 1), which reduces the query complexity to $O_\varepsilon(n + k^2)$.

3. **Accelerated Stochastic Greedy Improvement (Algorithm 2):** It can be shown that when the set $Z$ does not have a good value, it contains only little of the value of the optimal solution, and at the same time, it contains many of the elements that negatively affects this optimal solution. Thus, it makes sense to try to avoid this set. Accordingly, after obtaining the set $Z$, our algorithm constructs a second possible solution using a stochastic greedy algorithm that picks only elements of $\mathcal{N} \setminus Z$ in its first iterations. One can use for this purpose a version of the Random Greedy algorithm suggested by Buchbinder et al. [5] that uses $O(nk)$ queries (see Appendix A for details). To get the same result using fewer queries, we employ Algorithm 2 (described in Subsection 2.2), which is accelerated using ideas borrowed from the Sample Greedy algorithm of [6].

Our final algorithm (given as Algorithm 3 in Subsections 2.3) returns the better among the two sets produced in the last two steps (i.e., the output sets of Algorithm 1, and Algorithm 2). Intuitively, this algorithm guarantees our target approximation ratio of $0.385$ because when $f(Z)$ is smaller than this value, the set $Z$ is bad enough that avoiding it (in the first iterations) allows Algorithm 2 to get a good enough solution.

### 2.1 Fast local search

In this section, we present our accelerated local search algorithm, which is the algorithm used to implement the first two steps of our main algorithm. The properties of this algorithm are formally given by Theorem 2.1. Let $\mathbb{OPT}$ be an optimal solution.

**Theorem 2.1.** *There exists an algorithm that given a positive integer $k$, a value $\varepsilon \in (0,1)$, and a non-negative submodular function $f \colon 2^{\mathcal{N}} \to \mathbb{R}_{\geq 0}$, outputs a set $S \subseteq \mathcal{N}$ of size at max $k$ that, with probability at least $1 - \varepsilon$, obeys*

$$f(S) \geq \frac{f(S \cap \mathbb{OPT}) + f(S \cup \mathbb{OPT})}{2 + \varepsilon} \quad and \quad f(S) \geq \frac{f(S \cap \mathbb{OPT})}{1 + \varepsilon} \ .$$

*Furthermore, the query complexity of the above algorithm is $O_\varepsilon(n + k^2)$.*

Note that the guarantee of Theorem 2.1 is similar to the guarantee of a classical local search algorithm (see Appendix A for details). However, such a classical local search algorithm uses $O_\varepsilon(nk^2)$ queries, which is higher than the number of queries required for the algorithm from Theorem 2.1.

We defer the formal proof of Theorem 2.1 to Appendix B. However, we note that this proof is based on Algorithm 1. Algorithm 1 implicitly assumes that the ground set $\mathcal{N}$ includes at least $k + 1$ dummy

---

[1]A previous version of this paper used for this purpose the randomized Sample Greedy algorithm of [6]. Since this algorithm is randomized, to get a good solution with a high enough probability, that previous version had to run this algorithm $O(\log \varepsilon^{-1})$ times and select the solution with the highest function value. The code in the supplemental material of this paper includes the option to use either of these initialization options.

elements that always have a zero marginal contribution to $f$. Such elements can always be added to the ground set (before executing the algorithm) without affecting the properties of $f$, and removing them from the output set of the algorithm does not affect the guarantee of Theorem 2.1.

---

**Algorithm 1:** FAST-LOCAL-SEARCH$(k, f, \varepsilon, L)$

---

**input** : A positive integer $k \geq 1$, a submodular function $f$, an approximation factor $\varepsilon \in (0, 1)$, and a number $L$ of iterations.

**output :** A subset of $\mathcal{N}$ of cardinality at most $k$.

1 Initialize $S_0$ to be a feasible solution that with probability at least $1 - \varepsilon$ provides $c$-approximation for the problem for some constant $c \in (0, 1]$.

2 Fill $S_0$ with dummy elements to ensure $|S_0| = k$.

3 **for** $j = 1$ *to* $\lceil \log_2 \frac{1}{\varepsilon} \rceil$ **do**

4     Let $S_0^j \leftarrow S_0$.

5     **for** $i = 1$ *to* $L$ **do**

6        $Z_i^j \leftarrow$ Sample $\frac{n}{k}$ items from $\mathcal{N}$ uniformly at random.

7        $u_i^j \leftarrow \arg\max_{u' \in Z_i^j} f(u_i^j \mid S_{i-1}^j)$.

8        **if** $f(u_i^j \mid S_{i-1}^j) \leq 0$ **then** $u_i^j \leftarrow$ dummy element that does not belong to $S_{i-1}^j$.

9        $v_i^j \leftarrow \arg\min_{v' \in S_{i-1}^j} f(v' \mid S_{i-1}^j - v')$.

10       **if** $f(S_{i-1}^j) < f(S_{i-1}^j - v_i^j + u_i^j)$ **then** $S_i^j \leftarrow S_{i-1}^j - v_i^j + u_i^j$.

11       **else** $S_i^j \leftarrow S_{i-1}^j$.

12     Pick a uniformly random integer $0 \leq i^* < L$.

13     **if** *for every integer* $0 \leq t \leq k$ *it holds that*

$$\max_{S \subseteq \mathcal{N} \setminus S_{i^*}^j, |S| = t} \sum_{u \in S} f(u \mid S_{i^*}^j) \leq \min_{S \subseteq S_{i^*}^j, |S| = t} \sum_{v \in S} f(v \mid S_{i^*}^j - v) + \varepsilon f(S_{i^*}^j) \text{ **then return** } S_{i^*}^j.$$

14 **return** FAILURE.

---

Algorithm 1 starts by finding an initial solution $S_0$ guaranteeing constant approximation (we implement this step using the deterministic $1/4$-approximation algorithm of Balkanski et al. [2]). If the size of the initial solution is less than $k$ (i.e., $|S_0| < k$), the algorithm adds to it $k - |S_0|$ dummy elements. Then, Algorithm 1 makes roughly $\log_2 \varepsilon^{-1}$ attempts to find a good output. Each attempt tries to improve the (same) initial solution using $L$ iterations. Each iteration consisting of three steps: In Step (i), the algorithm samples $\frac{n}{k}$ items, and picks the element $u$ from the sample with the largest marginal contribution to the current solution $S_{i-1}$. If there are no elements in the sample with a positive marginal contribution, the algorithm picks a dummy element outside $S_{i-1}$ as $u$. In Step (ii), the algorithm picks the element $v \in S_{i-1}$ that has the lowest marginal value, i.e., the element whose removal from $S_{i-1}$ would lead to the smallest drop in value. In Step (iii), the algorithm swaps the elements $u$ and $v$ if such a swap increases the value of the current solution. Once $L$ iterations are over, the algorithm picks a uniformly random solution among all the solutions seen during this attempt (recall that the algorithm makes roughly $\log_2 \varepsilon^{-1}$ attempts to find a good solution). If the random solution found obeys the technical condition given on Line 13, then the algorithm returns it. Otherwise, the algorithm continues to the next attempt. If none of the attempts returns a set, the algorithm admits failure.

## 2.2 Guided stochastic greedy

In this section, we prove Theorem 2.2, which provides the last step of our main algorithm.

**Theorem 2.2.** *There exists an algorithm that given a positive integer $k$, a value $\varepsilon \in (0, 1)$, a value $t_s \in [0, 1]$, a non-negative submodular function $f : 2^{\mathcal{N}} \to \mathbb{R}_{\geq 0}$, and a set $Z \subseteq \mathcal{N}$ obeying the inequalities stated in Theorem 2.1, outputs a solution $S_k$, obeying*

$$\mathbb{E}[f(S_k)] \geq \Big( \frac{k - \lceil t_s \cdot k \rceil}{k} \alpha^{k - \lceil t_s \cdot k \rceil - 1} + \alpha^{k - \lceil t_s \cdot k \rceil} - \alpha^k \Big) f(\mathbb{OPT}) +$$

$$+ \Big( \alpha^k + \alpha^{k-1} - \frac{2k - \lceil t_s \cdot k \rceil}{k} \alpha^{k - \lceil t_s \cdot k \rceil - 1} \Big) f(\mathbb{OPT} \cup Z)$$

$$+ (\alpha^k - \alpha^{k - \lceil t_s \cdot k \rceil}) f(\mathbb{OPT} \cap Z) - 2\varepsilon f(\mathbb{OPT}) \ ,$$

*where $\alpha \triangleq 1 - 1/k$. Moreover, this algorithm requires only $O_\varepsilon(n)$ queries to the objective function.*

The algorithm used to prove Theorem 2.2 is Algorithm 2. This algorithm starts with an empty set and adds elements to it in iterations (at most one element per iteration) until its final solution is ready after $k$ iterations. In its first $\lceil k \cdot t_s \rceil$ iterations, the algorithm ignores the elements of $Z$, and in the other iterations, it considers all elements. However, except for this difference, the behavior of the algorithm in all iterations is very similar. Specifically, in each iteration $i$ the algorithm does the following two steps. In Step (i), the algorithm samples a subset $M_i$ containing $O_\varepsilon(n/k)$ elements from the data. In Step (ii), the algorithm considers a subset of $M_i$ (of size either $s_1 \lceil p(n - |Z|) \rceil$ or $s_2 \lceil pn \rceil$) containing the elements of $M_i$ with the largest marginal contributions with respect to the current solution $S_{i-1}$, and adds a uniformly random element out of this subset to the solution (if this element has a positive marginal contribution).

---

**Algorithm 2:** Guided Stochastic Greedy

---

**input** : A set $Z \subseteq \mathcal{N}$, a positive integer $k \geq 1$, values $\varepsilon \in (0, 1)$ and $t_s \in [0, 1]$, and a non-negative submodular function $f$

**output** : A set $S_k \subseteq \mathcal{N}$

**1** Initialize $S_0 \leftarrow \emptyset$.

**2** Define $p \leftarrow \min\{1, 8k^{-1}\varepsilon^{-2}\ln(2\varepsilon^{-1})\}$.

**3** Define $s_1 \leftarrow k/(n - |Z|)$ and $s_2 \leftarrow k/n$.

**4 for** $i = 1$ *to* $\lceil k \cdot t_s \rceil$ **do**

**5**  $\quad$ Let $M_i \subseteq \mathcal{N} \setminus Z$ be a uniformly random set containing $\lceil p \cdot (n - |Z|) \rceil$ elements.

**6**  $\quad$ Let $d_i$ be uniformly random scalar from the range $(0, s_1 \lceil p \cdot (n - |Z|) \rceil]$.

**7**  $\quad$ Let $u_i$ be an element of $M_i$ associated with the $\lceil d_i \rceil$-th largest marginal contribution to $S_{i-1}$ (if $\lceil d_i \rceil > |M_i|$, we set $u_i$ to be a dummy element having $0$ marginal contribution to $f$).

**8**  $\quad$ **if** $f(u_i \mid S_{i-1}) \geq 0$ **then**

**9**  $\quad\quad$ $S_i \leftarrow S_{i-1} \cup \{u_i\}$.

**10**  $\quad$ **else**

**11**  $\quad\quad$ $S_i \leftarrow S_{i-1}$.

**12 for** $i = \lceil k \cdot t_s \rceil + 1$ *to* $k$ **do**

**13**  $\quad$ Let $M_i \subseteq \mathcal{N}$ be a uniformly random set containing $\lceil p \cdot n \rceil$ elements.

**14**  $\quad$ Let $d_i$ be uniformly random scalar from the range $(0, s_2 \lceil p \cdot n \rceil]$.

**15**  $\quad$ Let $u_i$ be an element of $M_i$ associated with the $\lceil d_i \rceil$-th largest marginal contribution to $S_{i-1}$.

**16**  $\quad$ **if** $f(u_i \mid S_{i-1}) \geq 0$ **then** $S_i \leftarrow S_{i-1} \cup \{u_i\}$.

**17**  $\quad$ **else** $S_i \leftarrow S_{i-1}$.

**18 return** $S_k$.

---

### 2.3 $0.385$-Approximation guarantee

In this section, our objective is to prove the following theorem.

**Theorem 2.3.** *Given an integer $k \geq 1$ and a non-negative submodular function $f \colon 2^{\mathcal{N}} \to \mathbb{R}_{\geq 0}$, there exists an $0.385$-approximation algorithm for the problem of finding a set $S \subseteq \mathcal{N}$ of size at most $k$ maximizing $f$. This algorithm uses $O(n + k^2)$ queries to the objective function.*

The algorithm used to prove Theorem 2.3 is Algorithm 3. Our technical guarantee for Algorithm 3 is given as Lemma 2.4. When $k$ is large enough, this lemma immediately implies Theorem 2.3 by choosing $\varepsilon$ to be a small enough positive constant. If $k$ is small, getting Theorem 2.3 from Lemma 2.4 requires a three steps process. First, we choose an integer constant $\rho$ such that $\rho k$ is large enough, and we create a new ground set $\mathcal{N}_\rho = \{u_i \mid u \in \mathcal{N}, i \in [\rho]\}$ and a new objective function $g \colon 2^{\mathcal{N}_\rho} \to \mathbb{R}$ defined as $g(S) = \mathbb{E}[f(R(S))]$, where $R(S)$ is a random subset of $\mathcal{N}$ that includes every element $u \in \mathcal{N}$ with probability $|S \cap (\{u\} \times [\rho])|/\rho$. Then, we use Lemma 2.4 to get a set $\hat{S}$ that provides $0.385$-approximation for the problem $\max\{g(S) \mid |S| \leq \rho k\}$. Finally, the Pipage Rounding technique of [8] can be used to get from $\hat{S}$ a $0.385$-approximation for our original problem. Notice that since the size of $\hat{S}$ is constant (as we consider the case of a small $k$), this rounding can be done using a constant number of queries to the objective.

**Lemma 2.4.** *Algorithm 3 makes $O_\varepsilon(n + k^2)$ queries to the objective function, and returns a set whose expected value is at least $(c - O(\varepsilon + k^{-1}))f(\mathbb{OPT})$ for some constant $c > 0.385$.*

---

**Algorithm 3:** A 0.385-approximation algorithm for submodular maximization

---

**input** : A positive integer $k \geq 1$, a non-negative submodular function $f$, error parameter $\varepsilon \in (0, 1)$, and a flip point $0 \leq t_s \leq 1$

**output** : A set $S_L \subseteq \mathcal{N}$

**1** $Z \leftarrow$ FAST-LOCAL-SEARCH$(k, f, \varepsilon, L := \lceil 2k/(\varepsilon(1 - 1/e)) \rceil)$.

**2 if** *the last algorithm did not fail* **then**

**3** $\quad$ $A \leftarrow$ GUIDED-STOCHASTIC-GREEDY$(Z, k, t_s, \varepsilon)$ .

**4** $\quad$ **return** the set maximizing $f$ among $Z$ and $A$.

**5 else return** $\emptyset$.

---

*Proof.* According to the proof of Theorem 2.1, our choice of the parameter $L$ in Algorithm 1 guarantees that with probability at least $1 - \varepsilon$ the set $Z$ obeys the inequalities

$$f(Z) \geq \frac{f(Z \cup \mathbb{OPT}) + f(Z \cap \mathbb{OPT})}{2 + \varepsilon} \qquad \text{and} \qquad f(Z) \geq \frac{f(Z \cap \mathbb{OPT})}{1 + \varepsilon} \ .$$

Let us denote by $\mathcal{E}$ the event that these inequalities hold. By Theorem 2.2,

$$\mathbb{E}[f(A) \mid \mathcal{E}] \geq \Big(\frac{k - \lceil t_s \cdot k \rceil}{k}\alpha^{k - \lceil t_s \cdot k \rceil - 1} + \alpha^{k - \lceil t_s \cdot k \rceil} - \alpha^k \Big)f(\mathbb{OPT}) +$$

$$+ \Big(\alpha^k + \alpha^{k-1} - \frac{2k - \lceil t_s \cdot k \rceil}{k}\alpha^{k - \lceil t_s \cdot k \rceil - 1}\Big)\mathbb{E}[f(\mathbb{OPT} \cup Z) \mid \mathcal{E}]$$

$$+ (\alpha^k - \alpha^{k - \lceil t_s \cdot k \rceil})\mathbb{E}[f(\mathbb{OPT} \cap Z) \mid \mathcal{E}] - 2\varepsilon f(\mathbb{OPT}) \ .$$

Since the output of Algorithm 2 is the better set among $A$ and $Z$, we can lower bound its value by any convex combination of lower bounds on the values of $A$ and $Z$. More formally, if we denote by $p_1$, $p_2$ and $p_3$ any three non-negative values that add up to 1, then we get

$$\mathbb{E}[\max\{f(A), f(Z)\} \mid \mathcal{E}] \geq p_3 \Big(\frac{k - \lceil t_s \cdot k \rceil}{k}\alpha^{k - \lceil t_s \cdot k \rceil - 1} + \alpha^{k - \lceil t_s \cdot k \rceil} - \alpha^k \Big)f(\mathbb{OPT})$$

$$+ \Big(\frac{p_1}{2 + \varepsilon} + p_3\Big(\alpha^k + \alpha^{k-1} - \frac{2k - \lceil t_s \cdot k \rceil}{k}\alpha^{k - \lceil t_s \cdot k \rceil - 1}\Big)\Big)\mathbb{E}[f(\mathbb{OPT} \cup Z) \mid \mathcal{E}] \qquad (1)$$

$$+ \Big(\frac{p_2}{1 + \varepsilon} + \frac{p_1}{2 + \varepsilon} - p_3\Big(\alpha^{k - \lceil t_s \cdot k \rceil} - \alpha^k\Big)\Big)\mathbb{E}[f(\mathbb{OPT} \cap Z) \mid \mathcal{E}] - 2\varepsilon p_3 f(\mathbb{OPT}) \ .$$

To simplify the above inequality, we need to bound some of the terms in it. First,

$$\frac{k - \lceil t_s \cdot k \rceil}{k}\alpha^{k - \lceil t_s \cdot k \rceil - 1} + \alpha^{k - \lceil t_s \cdot k \rceil} - \alpha^k \geq \Big(2 - t_s - \frac{1}{k}\Big)\alpha^{k(1 - t_s)} - \alpha^k$$

$$\geq \Big(2 - t_s - \frac{1}{k}\Big)e^{t_s - 1}\Big(1 - \frac{1}{k}\Big)^{1 - t_s} - e^{-1}$$

$$\geq \Big(2 - t_s - \frac{3}{k}\Big)e^{t_s - 1} - e^{-1} \geq \Big(2 - t_s - e^{-t_s}\Big)e^{t_s - 1} - \frac{3}{k} \ ,$$

where the first inequality holds since $\lceil t_s \cdot k \rceil \leq t_s \cdot k + 1$, $\alpha \leq 1$, the second inequality follows since $\alpha^{k(1 - t_s)} \geq e^{t_s - 1}\big(1 - \frac{1}{k}\big)^{1 - t_s}$, and the last inequality holds since $e^{t_s - 1} \leq 1$. Second,

$$\alpha^k + \alpha^{k-1} - \frac{2k - \lceil t_s \cdot k \rceil}{k}\alpha^{k - \lceil t_s \cdot k \rceil - 1} \geq 2e^{-1} - \frac{2k - t_s \cdot k}{k}\alpha^{k - t_s \cdot k - 2} - \frac{2e^{-1}}{k}$$

$$\geq 2e^{-1} - (2 - t_s)e^{t_s - 1} - \frac{8 + 2e^{-1}}{k}$$

$$= -e^{t_s - 1}\big(2 - t_s - 2e^{-t_s}\big) - \frac{8 + 2e^{-1}}{k} \ ,$$

where the first inequality holds since $\alpha^{k-1} \geq \alpha^k \geq e^{-1}\big(1 - \frac{1}{k}\big)$, and the second inequality holds since $\alpha^{k - t_s \cdot k - 2} \leq e^{t_s - 1} + 4/k$. Finally, it holds that $\alpha^k - \alpha^{k - \lceil t_s \cdot k \rceil} \geq e^{-1}(1 - \frac{1}{k}) - e^{t_s - 1}/(1 - \frac{1}{k}) \geq e^{-1}(1 - \frac{1}{k}) - e^{t_s - 1}(1 + \frac{1}{k}) \geq -e^{t_s - 1}(1 - e^{-t_s}) - \frac{2}{k}$.

Plugging all the above lower bounds into Inequality (1) yields the promised simplified guarantee that

$$\mathbb{E}[\max\{f(A), f(Z)\} \mid \mathcal{E}] \geq p_3\big(2 - t_s - e^{-t_s}\big)e^{t_s - 1}f(\mathbb{OPT}) - O(\varepsilon + k^{-1})f(\mathbb{OPT})$$
$$+ \Big(\frac{p_1}{2 + \varepsilon} - p_3 e^{t_s - 1}\big(2 - t_s - 2e^{-t_s}\big) - O(k^{-1})\Big)\mathbb{E}[f(\mathbb{OPT} \cup Z) \mid \mathcal{E}]$$
$$+ \Big(\frac{p_2}{1 + \varepsilon} + \frac{p_1}{2 + \varepsilon} - p_3 e^{t_s - 1}\big(1 - e^{-t_s}\big) - O(k^{-1})\Big)\mathbb{E}[f(\mathbb{OPT} \cap Z) \mid \mathcal{E}] .$$

By [3], for an appropriate choice of values for $p_1$, $p_2$, $p_3$ and $t_s$ the last inequality implies

$$\mathbb{E}[\max\{f(A), f(Z)\} \mid \mathcal{E}] \geq (c - O(\varepsilon + k^{-1}))f(\mathbb{OPT})$$
$$- O(k^{-1}) \cdot \mathbb{E}[f(\mathbb{OPT} \cup Z) + f(\mathbb{OPT} \cap Z)] \quad (2)$$

for some constant $c > 0.385$. To get from the last inequality the bound on $\mathbb{E}[\max\{f(A), f(Z)\} \mid \mathcal{E}]$ stated in the lemma, we need to show that the conditioning on $\mathcal{E}$ and the last term of the inequality can both be dropped. To see why the conditioning can be dropped, note that the event $\mathcal{E}$ happens with probability at least $1 - \varepsilon$, and when it does not happen the set returned by the algorithm still has a non-negative value. These observations show together that removing the conditioning on $\mathcal{E}$ in Inequality (2) only affects the constant inside the big $O$ notation. Notice now that since $\mathbb{OPT} \cap Z \subseteq \mathbb{OPT}$ is always a feasible solution, it deterministically holds that $f(\mathbb{OPT} \cap Z) \leq f(\mathbb{OPT})$. Similarly, since $Z$ is a feasible solution, the submodularity of $f$ guarantees that $f(\mathbb{OPT} \cup Z) + f(\mathbb{OPT} \cap Z) \leq f(\mathbb{OPT}) + f(Z) \leq 2f(\mathbb{OPT})$. These two bounds allow us to drop the last term of Inequality (2) at the cost of increasing (again) the constant inside the big $O$ notation.

To complete the proof of the lemma, note that Line 1 of Algorithm 3 requires $O_\varepsilon\big(n + k^2\big)$ queries to the objective function as shown in the proof of Theorem 2.1, while Line 3 of Algorithm 3 requires $O_\varepsilon(n)$ queries to the objective function as dictated by Theorem 2.2. $\qquad\square$

## 3 Experiments

To emphasize the effectiveness of our suggested method from Section 2, in this section, we empirically compare Algorithm 3 with two benchmark algorithms on three machine-learning applications: movie recommendation, image summarization, and revenue maximization. Each one of these applications necessitates maximization of a non-monotone submodular function. The benchmark algorithms we consider are the Random Greedy algorithm of Buchbinder et al. [5], and the Random Sampling algorithm of [6]. These algorithms are the current state-of-the-art practical algorithms for maximizing non-monotone submodular functions.

As stated, Algorithm 2 requires $O\big(\frac{n \cdot \ln \varepsilon^{-1}}{\varepsilon^2}\big)$ queries to the objective function, where the dependence on $\varepsilon$ comes from the choice of value for the parameter $p$ of the algorithm. However, we have found out that in practice a more modest choice of value for $p$ suffices. Specifically, in our experiments, we have replaced Line 2 of Algorithm 2 with $p \leftarrow \min\{1, \frac{8}{k \cdot \varepsilon}\}$. Throughout the experiments, we have set $\varepsilon = 0.1$; and all the reported results are averaged across 8 executions. We use shades in our plots to depict the standard deviations of the individual results obtained in these 8 executions.

**Software/Hardware**. Our algorithms were implemented in Python 3.11 using mainly "Numpy" [19], and Numba [23]. The implementations' code can be found at `https://github.com/muradtuk/385ApproximationSubMax`. The experiments were performed on a 2.2GHz i9-13980HX (24 cores total) machine with 64GB RAM.

### 3.1 Personalized movie recommendation

Consider a movie recommendation system in which each user specifies what genres they are interested in, and the system has to provide a representative subset of movies from these genres. Assume that each movie is represented by a vector consisting of users' ratings for the corresponding movie. One challenge here is that each user does not necessarily rate all the movies. Hence, the vectors representing the movies do not necessarily have similar sizes. To overcome this challenge, low-rank matrix completion techniques [9] can be performed on the matrix with missing values to obtain a complete rating matrix. Formally, given a few ratings from $k$ users to $n$ movies we obtain in this way a rating matrix $\mathbf{M}$ of size $k \times n$. Following [33, 30], to score the quality of a selected subset of

movies, we use the function $f(S) = \sum_{u \in \mathcal{N}} \sum_{v \in S} s_{u,v} - \lambda \sum_{u \in S} \sum_{v \in S} s_{u,v}$. Here, $\mathcal{N}$ is the set of $n$ movies, $\lambda \in [0,1]$ is a parameter and $s_{u,v}$ denotes the similarity between movies $u$ and $v$ (the similarity $s_{u,v}$ can be calculated based on the matrix $\mathbf{M}$ in multiple ways: cosine similarity, inner product, etc). Note that the first term in $f$'s definition captures the coverage, while the second term captures diversity. Thus, the parameter $\lambda$ controls the importance of diversity in the returned subset. For any $\lambda \leq 0.5$, $f(S)$ is monotone [34], however, it can be non-monotone for larger values of $\lambda$.

We followed the experimental setup of the prior works [33, 30] and used a subset of movies from the MovieLens data set [18] which includes $10{,}437$ movies. Each movie in this data set is represented by a 25 dimensional feature vector calculated using user ratings, and we used the inner product similarity to obtain the similarity values $s_{u,v}$ based on these vectors. When experimenting with this application, we fixed $\lambda$ to be either $0.55$ or $0.75$, and varied $k$.

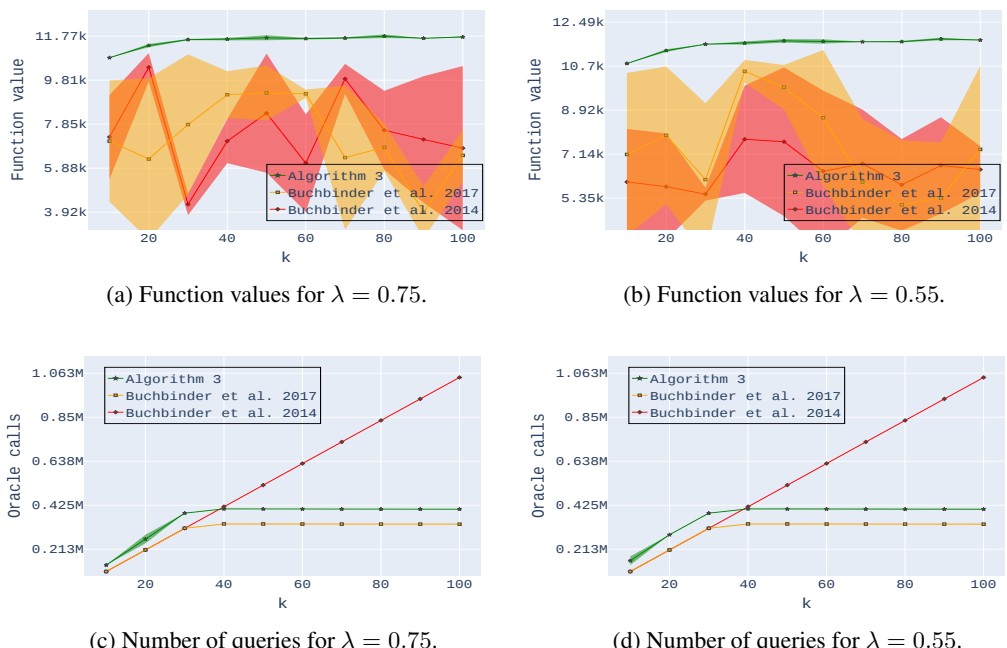

(a) Function values for $\lambda = 0.75$.

(b) Function values for $\lambda = 0.55$.

(c) Number of queries for $\lambda = 0.75$.

(d) Number of queries for $\lambda = 0.55$.

Figure 1: Experimental results for Personalized Movie Recommendation. Plots (a) and (b) compare the output of our algorithm with the benchmark algorithms mentioned at the beginning of Section 3 for a particular value of the parameter $\lambda$ and a varying number $k$ of movies. Plots (c) and (d) compare the number of queries used by the various algorithms.

The results of these experiments are depicted in Figure 1. One can observe that our proposed method, Algorithm 3, demonstrates superior performance compared to the other methods. Moreover, this performance is stable, and presents a much smaller variance compared to the variance in the performance of the two benchmark algorithms. The number of queries used by our algorithm is only slightly larger than the number of queries used by the Random Sampling algorithm of [6], and is typically smaller than the number of queries used by the Random Greedy algorithm of Buchbinder et al. [5], sometimes by as much as a factor of 2.

## 3.2 Personalized image summarization

Consider a setting in which we get as input a collection $\mathcal{N}$ of images from $\ell$ disjoint categories (e.g., birds, dogs, cats) and the user specifies $r \in [\ell]$ categories, and then demands a subset of the images in these categories that summarizes all the images of the categories. Following [30] again, to evaluate a given subset of images, we use the function $f(S) = \sum_{u \in \mathcal{N}} \max_{v \in S} s_{u,v} - \frac{1}{|\mathcal{N}|} \sum_{u \in S} \sum_{v \in S} s_{u,v}$, where $s_{u,v}$ is a non-negative similarity between images $u$ and $v$.

To obtain the similarity between pair of images $u, v$, we utilized the *DINO-VITB16* model [10] from HuggingFace [44] as the feature encoder for vision datasets. Specifically, the final layer CLS token

embedding output was used as the feature representation. The similarity between pairs of images was then computed as the cosine similarity of the corresponding embedding vectors. To experiment in this setting, we used three datasets: (i) *CIFAR10* [21] – A dataset of 50,000 images belonging to 10 different classes (categories). (ii) *CIFAR100* [21] – A dataset of 50,000 images belonging to 100 different classes. (iii) *Tiny ImageNet* [24] – A dataset of 100,000 images belonging to 200 different classes. In each one of our experiments, the task was to summarize a set of 10,000 images sampled uniformly from one of these datasets. The upper bound $k$ on the number images allowed in the summary varied between experiments. The results of our experiments are depicted in Figure 2.

Similarly to Section 3.1, we observe that our algorithm (Algorithm 3) produces higher values compared to the state-of-the-art practical algorithms, and enjoys a lower variance in the quality of its output. However, due to the small values used for $k$, our algorithm requires significantly more queries to the objective function compared to the two other algorithms.

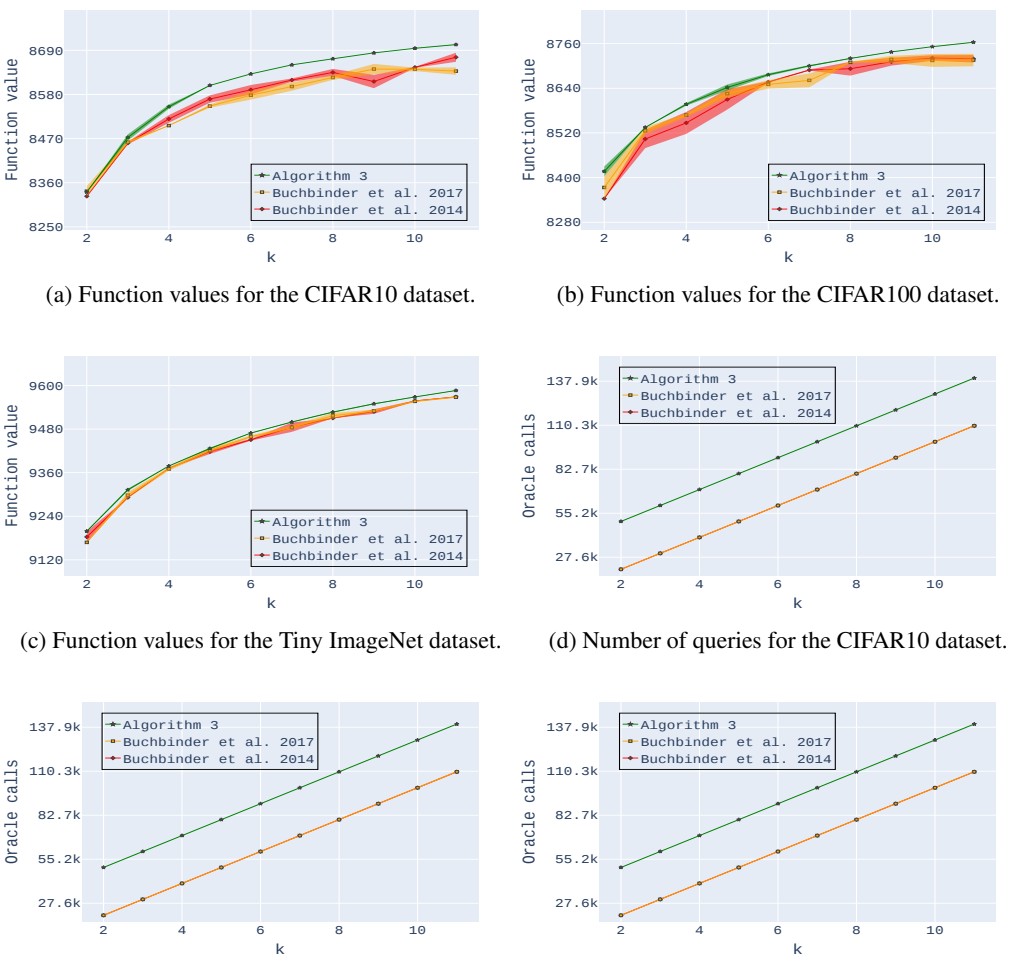

(a) Function values for the CIFAR10 dataset.

(b) Function values for the CIFAR100 dataset.

(c) Function values for the Tiny ImageNet dataset.

(d) Number of queries for the CIFAR10 dataset.

(e) Number of queries for the CIFAR100 dataset.

(f) Number of queries for the Tiny ImageNet dataset.

Figure 2: Experimental results for Personalized Image Summarization. Plots (a)–(c) compare the output of our algorithm with the benchmark algorithms mentioned at the beginning of Section 3 for a varying number $k$ of images. Each plot corresponds to a different dataset. Plots (d)–(f) compare the number of queries used by the various algorithms.

## 3.3 Revenue maximization

Consider a company whose objective is to promote a product to users to boost revenue through the "word-of-mouth" effect. More specifically, given a social network, we need to choose a subset of

up to $k$ users to receive a product for free in exchange for advertising it to their network neighbors, and the goal is to choose users in a manner that maximizes revenue. The problem of optimizing this objective can be formalized as follows. The input is a weighted undirected graph $G = (V, E)$ representing a social network, where $w_{ij}$ represents the weight of the edge between vertex $i$ and vertex $j$ (with $w_{ij} = 0$ if the edge $(i, j)$ is absent from the graph). Given a set $S \subseteq V$ of users who have become advocates for the product, the expected revenue generated is proportional to the total influence of $S$'s users on non-advocate users, formally expressed as $f(x) = \sum_{i \in S} \sum_{j \in V \setminus S} w_{ij}$. It has been demonstrated that $f$ is non-monotone and submodular [1].

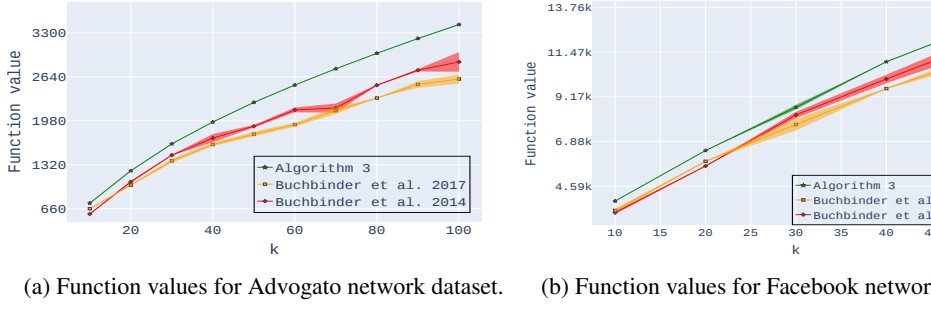

(a) Function values for Advogato network dataset.     (b) Function values for Facebook network dataset.

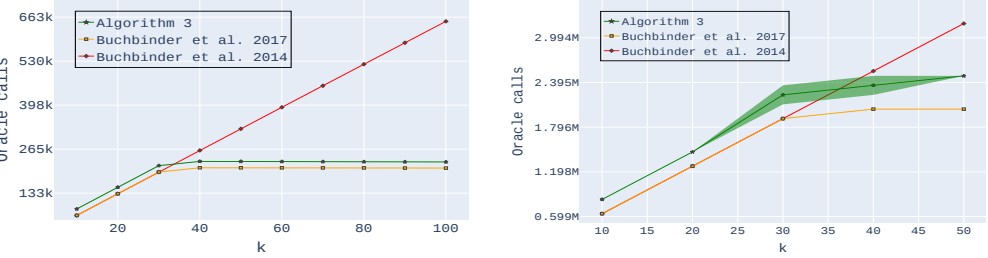

(c) Number of queries for Advogato network dataset. (d) Number of queries for Facebook network dataset.

Figure 3: Experimental results for Revenue Maximization. Plots (a) and (b) compare the output of our algorithm with the benchmark algorithms mentioned at the beginning of Section 3 for a varying number $k$ of images on the Advogato and Facebook network datasets. Plots (c) and (d) compare the number of queries used by the various algorithms.

In our experiment, we compared the performance of Algorithm 3 and the two benchmark algorithms on the Facebook network [42] and the Advogato network [29]. The results of this experiment are depicted in Figure 3. Once again, our algorithm enjoys both better output values and lower standard deviations compared to the benchmark algorithms. Our algorithm uses more queries compared to the Random Sampling algorithm of [6], but the ratio between the number of queries used by the two algorithms tends to decrease as $k$ increases. The behavior of the Random Greedy algorithm of [5] greatly depends on $k$. For smaller values of $k$ this algorithm requires roughly as many queries as Random Sampling, but for larger value of $k$ it requires significantly more queries than our algorithm.

## 4   Conclusion

In this work, we have presented a novel algorithm for submodular maximization subject to cardinality constraint that combines a practical query complexity of $O(n + k^2)$ with an approximation guarantee of $0.385$, which improves over the $1/e$-approximation of the state-of-the-art practical algorithms. In addition to giving a theoretical analysis of our algorithm, we have demonstrated its empirical superiority (compared to practical state-of-the-art methods) in various machine learning applications. We hope future work will be able to improve the query complexity of our algorithm to be cleanly linear without sacrificing either the approximation guarantee or the practicality of the algorithm.

## Acknowledgment

The work of Loay Mualem and Moran Feldman was supported in part by Israel Science Foundation (ISF) grant number 459/20.

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

# A A warmup version of our algorithm

In this section, we present and analyze a simpler version of our algorithm with the same general structure, but excluding the speedup techniques used to obtain our main result. Inspired by Buchbinder et al. [3], and similar to Algorithm 3, this simpler version (given as Algorithm 6) comprises three steps: (i) Searching for a good initial solution that guarantees a constant approximation to the optimal set. This is accomplished by running the Twin Greedy algorithm of [17], (ii) Finding an (approximate) local search optimum set $Z$ using a local search method. In this simple version, we use the classical local search algorithm for this purpose, which requires $O(nk^2)$ queries to the objective function. (iii) Lastly, we construct another solution using a version of the Random Greedy algorithm suggested by Buchbinder et al. [5] that avoids elements of the set $Z$ in its first iterations. The algorithm terminates by outputting the better of the two solutions generated in the last two steps.

## A.1 Local search

In this section, we present (as Algorithm 4) a simple local search algorithm, which is the algorithm used to implement the first two steps of Algorithm 6. Algorithm 4 begins by finding an initial solution $S$ using the Twin Greedy algorithm of [17]. The algorithm then proceeds as follows: (i) If $|S| < k$, it checks for an element $u$ such that adding $u$ to $S$ increases the function value by at least $(1 + \frac{\varepsilon}{2k})f(S)$. If such an element is found, it is added to $S$. (ii) If $|S| = k$, it looks for two elements $u \in \mathcal{N} \setminus S$ and $v \in S$ such that swapping $u$ and $v$ (i.e., removing $v$ from $S$ and adding $u$ to $S$) increases the function value by at least $(1 + \frac{\varepsilon}{2k})f(S)$. If such elements exist, the algorithm performs the swap. (iii) If no elements satisfy the previous two conditions, the algorithm checks for an element $v$ such that removing $v$ from $S$ increases the function value by $(1 + \frac{\varepsilon}{2k})f(S)$. If such an element is found, it is removed from $S$. Algorithm 4 continues to search for elements satisfying any of the above three conditions until no such elements exist any longer. When this happens, the algorithm terminates and returns the set $S$ as its output.

---

**Algorithm 4:** LOCAL-SEARCH$(k, f)$

**input** : A positive integer $k \geq 1$, a non-negative submodular function $f$, and an error parameter $\varepsilon \in (0, 1)$.
**output** : A set $S \subseteq \mathcal{N}$

1 Initialize $S$ to be a feasible solution guaranteeing $c$-approximation for the problem for some constant $c \in (0, 1]$.
2 **while** true **do**
3     **if** $\exists u \in \mathcal{N} \setminus S$ *such that* $f(S + u) \geq \left(1 + \frac{\varepsilon}{2k}\right)f(S)$ *and* $|S| < k$ **then**
4         $S \leftarrow S + u$.
5     **else if** $\exists u \in \mathcal{N} \setminus S, v \in S$ *such that* $f(S + u - v) \geq \left(1 + \frac{\varepsilon}{2k}\right)f(S)$ *and* $|S| = k$ **then**
6         $S \leftarrow S - v + u$.
7     **else if** $\exists v \in S$ *such that* $f(S - v) \geq \left(1 + \frac{\varepsilon}{2k}\right)f(S)$ **then**
8         $S \leftarrow S - v$.
9     **else**
10         **return** S

---

The properties of Algorithm 4 are formally established by Theorem A.1.

**Theorem A.1.** *Given a positive integer $k$, a non-negative submodular function $f$ and an error parameter $\varepsilon \in (0, 1)$, Algorithm 4 returns a set $S \subseteq \mathcal{N}$ of size at most $k$ such that*

$$f(S) \geq \frac{f(S \cup \mathbb{OPT}) + f(S \cap \mathbb{OPT})}{2 + \varepsilon} \quad \text{and} \quad \frac{f(S \cap \mathbb{OPT})}{1 + \varepsilon} \; ,$$

*while requiring $O_\varepsilon\left(nk^2\right)$ queries to the objective function.*

*Proof.* Let $S$ denote the output of Algorithm 4, and let $E_-$, $E_\pm$, and $E_+$ be defined as follows.

- $E_-$ is the event that there exists $v \in S$ such that $f(S - v) \geq \left(1 + \frac{\varepsilon}{2k}\right)f(S)$.

- $E_{\pm}$ is the event that there exists $u \in \mathcal{N} \setminus S, v \in S$ such that $f(S + u - v) \geq \left(1 + \frac{\varepsilon}{2k}\right) f(S)$ and $|S| = k$.

- $E_+$ is the event that there exists $u \in \mathcal{N} \setminus S$ such that $f(S + u) \geq \left(1 + \frac{\varepsilon}{2k}\right) f(S)$ and $|S| < k$.

Since the algorithm terminated with the set $S$, none of the events $E_+$, $E_-$ or $E_{\pm}$ occurs. To lower bound the value of the set $S$, we inspect the implications arising from each of these events not occurring.

**Implications of $E_-$ not occurring.** Since $E_-$ does not occur, for every $v \in S$, $\left(1 + \frac{\varepsilon}{2k}\right) f(S) > f(S - v)$. Summing this inequality across every element in $v \in S \setminus \mathbb{OPT}$ yields

$$\left(1 + \frac{\varepsilon}{2k}\right) f(S) \geq f(S) + \frac{1}{|S \setminus \mathbb{OPT}|} \sum_{v \in S \setminus \mathbb{OPT}} [f(S - v) - f(S)]$$

$$\geq f(S) + \frac{1}{|S \setminus \mathbb{OPT}|} (f(S \cap \mathbb{OPT}) - f(S)) \;,$$

where the second inequality holds by submodularity of $f$. By rearrangement, we obtain that

$$f(S) \geq \frac{1}{\frac{\varepsilon}{2k}|S \setminus \mathbb{OPT}| + 1} f(S \cap \mathbb{OPT}) \geq \frac{1}{1 + \varepsilon/2} f(S \cap \mathbb{OPT}) \;, \tag{3}$$

where the last inequality holds since $|S \setminus \mathbb{OPT}| \leq |S| \leq k$.

The above proves the second inequality guaranteed by the theorem. Below we show that the first inequality guaranteed by the theorem is implied either by $E_+$ not occurring, or by $E_{\pm}$ not occuring, depending on the size of $S$.

**Implications of $E_+$ not occurring when $|S| < k$.** Since $E_+$ does not occur and $|S| < k$, for every $u \in \mathcal{N} \setminus S$, it holds that $\left(1 + \frac{\varepsilon}{2k}\right) f(S) > f(S + u)$. Summing the above inequality across every element in $u \in \mathbb{OPT} \setminus S$ yields

$$\left(1 + \frac{\varepsilon}{2k}\right) f(S) \geq f(S) + \frac{1}{|\mathbb{OPT} \setminus S|} \sum_{u \in \mathbb{OPT} \setminus S} [f(S + u) - f(S)]$$

$$\geq f(S) + \frac{1}{|\mathbb{OPT} \setminus S|} (f(S \cup \mathbb{OPT}) - f(S)) \;,$$

where the second inequality follows from the submodularity of $f$. By rearrangement, we now obtain

$$f(S) \geq \frac{1}{\frac{\varepsilon}{2k}|\mathbb{OPT} \setminus S| + 1} f(S \cup \mathbb{OPT}) \geq \frac{1}{1 + \varepsilon/2} f(S \cup \mathbb{OPT}) \;, \tag{4}$$

where the last inequality holds since $|\mathbb{OPT} \setminus S| \leq |\mathbb{OPT}| \leq k$. Averaging Inequalities (3) and (4) gives

$$f(S) \geq \frac{f(S \cup \mathbb{OPT}) + f(S \cap \mathbb{OPT})}{2 + 2\varepsilon} \;,$$

which proves the first inequality guaranteed by the theorem in the case of $|S| < k$.

**Implications of $E_{\pm}$ not occurring when $|S| = k$.** Since $E_{\pm}$ does not occur and $|S| = k$, for every pair $u, v$ where $u \in \mathcal{N} \setminus S$ and $v \in S$, $\left(1 + \frac{\varepsilon}{2k}\right) f(S) \geq f(S + u - v)$. Summing the above inequality for every $v \in S \setminus \mathbb{OPT}$ and $u \in \mathbb{OPT} \setminus S$, yields that

$$\left(1 + \frac{\varepsilon}{2k}\right) f(S) \geq \frac{1}{|S \setminus \mathbb{OPT}| \cdot |\mathbb{OPT} \setminus S|} \sum_{u \in \mathbb{OPT} \setminus S} \sum_{v \in S \setminus \mathbb{OPT}} f(S + u - v) \;.$$

Observe that

$$
\frac{1}{|S \setminus \mathbb{OPT}| \cdot |\mathbb{OPT} \setminus S|} \sum_{u \in \mathbb{OPT} \setminus S} \sum_{v \in S \setminus \mathbb{OPT}} f(S + u - v) - f(S)
$$

$$
= \frac{\overbrace{\sum_{u \in \mathbb{OPT} \setminus S} \sum_{v \in S \setminus \mathbb{OPT}} [f(S + u - v) - f(S - v)]}^{\mathbb{A}}}{|S \setminus \mathbb{OPT}| \cdot |\mathbb{OPT} \setminus S|} + \frac{\overbrace{\sum_{u \in \mathbb{OPT} \setminus S} \sum_{v \in S \setminus \mathbb{OPT}} [f(S - v) - f(S)]}^{\mathbb{B}}}{|S \setminus \mathbb{OPT}| \cdot |\mathbb{OPT} \setminus S|} \ .
$$

To bound $\mathbb{A}$, note that, by the submodularity of $f$,

$$
\sum_{u \in \mathbb{OPT} \setminus S} [f(S + u - v) - f(S - v)] \geq \sum_{u \in \mathbb{OPT} \setminus S} [f(S + u) - f(S)]
$$
$$
\geq f(S \cup \mathbb{OPT}) - f(S) \ ,
$$

and hence,

$$
\frac{\sum_{u \in \mathbb{OPT} \setminus S} \sum_{v \in S \setminus \mathbb{OPT}} [f(S + u - v) - f(S - v)]}{|S \setminus \mathbb{OPT}| \cdot |\mathbb{OPT} \setminus S|} \geq \frac{f(S \cup \mathbb{OPT}) - f(S)}{|\mathbb{OPT} \setminus S|} \ .
$$

To bound $\mathbb{B}$, we note that the submodularity of $f$ implies that

$$
\frac{\sum_{u \in \mathbb{OPT} \setminus S} \sum_{v \in S \setminus \mathbb{OPT}} [f(S - v) - f(S)]}{|S \setminus \mathbb{OPT}| \cdot |\mathbb{OPT} \setminus S|} = \frac{\sum_{v \in S \setminus \mathbb{OPT}} [f(S - v) - f(S)]}{|S \setminus \mathbb{OPT}|} \geq \frac{f(S \cap \mathbb{OPT}) - f(S)}{|S \setminus \mathbb{OPT}|} \ .
$$

Combining all of the above yields that

$$
\frac{\varepsilon}{2k} f(S) \geq \frac{f(S \cap \mathbb{OPT}) - f(S)}{|S \setminus \mathbb{OPT}|} + \frac{f(S \cup \mathbb{OPT}) - f(S)}{|\mathbb{OPT} \setminus S|} \ ,
$$

which by rearrangement implies

$$
f(S) \geq \frac{f(S \cap \mathbb{OPT})}{1 + \frac{\varepsilon}{2k}|S \setminus \mathbb{OPT}| + \frac{|S \setminus \mathbb{OPT}|}{|\mathbb{OPT} \setminus S|}} + \frac{f(S \cup \mathbb{OPT})}{1 + \frac{\varepsilon}{2k}|\mathbb{OPT} \setminus S| + \frac{|\mathbb{OPT} \setminus S|}{|S \setminus \mathbb{OPT}|}}
$$
$$
\geq \frac{f(S \cup \mathbb{OPT}) + f(S \cap \mathbb{OPT})}{2 + \varepsilon},
$$

where the last inequality holds since $|\mathbb{OPT} \setminus S| = |S \setminus \mathbb{OPT}| \leq k$. This completes the proof of the first inequality guaranteed by the theorem in the case of $|S| = k$.

To complete the proof of the theorem, it remains to analyze the query complexity of Algorithm 1. First, we recall that $S$ is initialized on Line 1 of Algorithm 4 by the Twin Greedy algorithm of [17], which gives an approximation ratio $c = 1/5$ using $O(n \log k)$ queries to the objective function. Each iteration of the loop of Algorithm 4 can be implemented using $O(nk)$ queries since there are only $O(nk)$ ways to choose $u \in \mathcal{N} \setminus S$ and $v \in S$. Let us bound the number $L$ of such iterations. Observe that the function value of the set $S$ in Algorithm 4 increases by a multiplicative factor of $1 + \frac{\varepsilon}{2k}$ following each iteration of the while loop (except for the last one). Since $S$ is initialized with a solution of value at least $cf(\mathbb{OPT})$, and its value is never larger than $f(OPT)$ (because it remains feasible), we get

$$
f(OPT) \geq \left(1 + \frac{\varepsilon}{k}\right)^{L-1} cf(\mathbb{OPT}) \ ,
$$

and rearranging gives us

$$
L \leq 1 + \frac{\ln \frac{1}{c}}{\ln(1 + \frac{\varepsilon}{2k})} = 1 + \frac{\ln 5}{\ln(1 + \frac{\varepsilon}{2k})} = O(k/\varepsilon) \ .
$$

Combining the above results, we get that the query complexity of Algorithm 4 is upper bounded by $O(n \log k) + O(nk) \cdot L = O(n \log k) + O(nk) \cdot O(k/\varepsilon) = O(nk^2 \varepsilon^{-1})$. □

## A.2 Guided Random Greedy

In this section, we present the Guided Random Greedy algorithm (Algorithm 5), which is the variant of the Random Greedy algorithm of [5] used to implement the last step of Algorithm 6. Algorithm 5 starts with an empty set, and adds to it one element in each iteration until returning the final solution after $k$ iterations. In its first $\lceil k \cdot t_s \rceil$ iterations, the algorithm ignores the elements of $Z$, and in the rest of the iterations, it considers all elements. However, except for this difference, the behavior of the algorithm in all iterations is very similar. Specifically, in each iteration $i$ the algorithm does the following two steps. In Step (i) the algorithm finds a subset $M_i$ of size $k$ maximizing the sum of marginal gains of the elements $u \in M_i$ with respect to the current solution $S_{i-1}$. In step $(ii)$, the algorithm chooses a random element from $M_i$ and adds it to the solution. This algorithm implicitly assumes that $|\mathcal{N}| \geq 3k$. If this is not the case, one can fix that by adding to the ground set $2k$ dummy elements of value $0$ before executing the algorithm (and then removing any dummy elements that appear in the solution of the algorithm).

---

**Algorithm 5:** Guided Random Greedy

**input** : A set $Z \subseteq \mathcal{N}$, a positive integer $k \geq 1$, a non-negative submodular function $f$, and a flip point $t_s \in [0, 1]$
**output** : A set $S \subseteq \mathcal{N}$

1 Initialize $S_0 \leftarrow \emptyset$.
2 **for** $i = 1$ *to* $\lceil k \cdot t_s \rceil$ **do**
3      Let $M_i \subseteq \mathcal{N} \setminus (S_{i-1} \cup Z)$ be a subset of size $k$ maximizing $\sum_{u \in M_i} f(u \mid S_{i-1} + u)$.
4      Let $u_i$ be a uniformly random element from $M_i$.
5      $S_i \leftarrow S_{i-1} + u_i$.
6 **for** $i = \lceil k \cdot t_s \rceil + 1$ *to* $k$ **do**
7      Let $M_i \subseteq \mathcal{N} \setminus S_{i-1}$ be a subset of size $k$ maximizing $\sum_{u \in M_i} f(u \mid S_{i-1} + u)$.
8      Let $u_i$ be a uniformly random element from $M_i$.
9      $S_i \leftarrow S_{i-1} + u_i$.
10 **return** $S_k$.

---

The properties of Algorithm 5 are given by Theorem A.2.

**Theorem A.2.** *There exists an algorithm that given a positive integer $k$, a value $t_s \in [0, 1]$, a non-negative submodular function $f \colon 2^{\mathcal{N}} \to \mathbb{R}_{\geq 0}$, and a set $Z \subseteq \mathcal{N}$ obeying the inequalities given in Theorem A.1, outputs a solution $S_k$, obeying*

$$\mathbb{E}[f(S_k)] \geq \left( \frac{k - \lceil t_s \cdot k \rceil}{k} \alpha^{k - \lceil t_s \cdot k \rceil - 1} + \alpha^{k - \lceil t_s \cdot k \rceil} - \alpha^k \right) f(\mathbb{OPT}) +$$

$$+ \left( \alpha^k + \alpha^{k-1} - \frac{2k - \lceil t_s \cdot k \rceil}{k} \alpha^{k - \lceil t_s \cdot k \rceil - 1} \right) f(\mathbb{OPT} \cup Z)$$

$$+ \left( \alpha^k - \alpha^{k - \lceil t_s \cdot k \rceil} \right) f(\mathbb{OPT} \cap Z) \ ,$$

*where $\alpha = 1 - 1/k$. Furthermore, this algorithm requires only $O(nk)$ queries to the objective function.*

To prove Theorem A.2, we first need to present some preliminaries. The Lovász extension of $f$ is a function $\hat{f} \colon [0, 1]^{\mathcal{N}} \to \mathbb{R}$ defined as follows. For every vector $x \in [0, 1]^{\mathcal{N}}$,

$$\hat{f}(x) = \int_0^1 f(T_\lambda(x)) d\lambda \ ,$$

where $T_\lambda(x) \triangleq \{u \in \mathcal{N} \mid x_u \geq \lambda\}$. The Lovász extension of a submodular function is known to be convex. More important for us is the following known lemma regarding this extension. This lemma stems from an equality, proved by Lovász [28], between the Lovász extension of a submodular function and another extension known as the convex closure.

**Lemma A.3.** *Let $f \colon 2^{\mathcal{N}} \to \mathbb{R}$ be a submodular function, and let $\hat{f}$ be its Lovász extension. For every $\mathbf{x} \in [0, 1]^{\mathcal{N}}$ and random set $D_x \subseteq \mathcal{N}$ obeying $\Pr[u \in D_x] = x_u$ for every $u \in \mathcal{N}$ (i.e., the marginals of $D_x$ agree with $x$), $\hat{f}(x) \leq \mathbb{E}[f(D_x)]$.*

Using the last lemma, we now prove a lower bound on the expected value of the union of any set $A$ with the solution $S_i$ of Algorithm 5 after $i$ iterations.

**Lemma A.4.** *For every integer $0 \le i \le k$ and set $A \subseteq \mathcal{N}$, it holds that*

$$\mathbb{E}[f(S_i \cup A)] \ge \left(1 - \frac{1}{k}\right)^{\beta_i} \cdot f(A) - \left[\left(1 - \frac{1}{k}\right)^{\beta_i} - \left(1 - \frac{1}{k}\right)^{i-1}\right] \cdot f(A \cup Z) \;,$$

*where $\beta_i = \max\{0, i - \lceil t_s \cdot k \rceil\}$.*

*Proof.* Let $x^{(i)} \in [0, 1]^{\mathcal{N}}$ be the vector of the marginal probabilities of elements to belong to $S_i$. In other words, for every element $u \in \mathcal{N}$, $x_u^{(i)} = \Pr[u \in S_i]$. Since each iteration of Algorithm 5 adds each element to the solution with probability at most $1/k$, the coordinates of $x^{(i)}$ are all upper bounded by $1 - (1 - 1/k)^i$. For elements of $Z$ we also know that they are not added by Algorithm 5 to the solution in the first $\lceil k \cdot t_s \rceil$ iterations, and therefore, their coordinates in $x^{(i)}$ are upper bounded by

$$1 - (1 - 1/k)^{\max\{0, i - \lceil k \cdot t_s \rceil\}} = 1 - (1 - 1/k)^{\beta_i} \;.$$

Let $\mathbf{1}_A$ denote a vector in $\{0, 1\}^{\mathcal{N}}$ containing 1s at the entries that correspond to elements present in $A$ and 0 in the remaining coordinates. We also denote by $x^{(i)} \vee \mathbf{1}_A$ the coordinate-wise maximum of $x^{(i)}$ and $\mathbf{1}_A$. By Lemma A.3,

$$\mathbb{E}[f(S_i \cup A)] \ge \hat{f}(\mathbf{x}_{S_i} \vee \mathbf{1}_A) = \int_0^1 f(T_\lambda(\mathbf{x}_{S_i} \vee \mathbf{1}_A)) d\lambda$$

$$\ge \int_{1-(1-\frac{1}{k})^{\beta_i}}^{1-(1-\frac{1}{k})^{i-1}} f(T_\lambda(\mathbf{x}_{S_i} \vee \mathbf{1}_A)) d\lambda + \int_{1-(1-\frac{1}{k})^{i-1}}^{1} f(T_\lambda(\mathbf{x}_{S_i} \vee \mathbf{1}_A)) d\lambda$$

$$= \int_{1-(1-\frac{1}{k})^{\beta_i}}^{1-(1-\frac{1}{k})^{i-1}} f(T_\lambda(\mathbf{x}_{S_i} \vee \mathbf{1}_A)) d\lambda + \left(1 - \frac{1}{k}\right)^{i-1} f(A)$$

$$\ge \left[\left(1 - \frac{1}{k}\right)^{\beta_i} - \left(1 - \frac{1}{k}\right)^{i-1}\right] \cdot [f(A) - f(A \cup Z)] + \left(1 - \frac{1}{k}\right)^{i-1} f(A) \;,$$

where the second inequality holds by the non-negativity of $f$, and the last inequality follows since $T_\lambda(\mathbf{x}_{S_i} \vee \mathbf{1}_A) = B_\lambda \cup A$ for some set $B_\lambda \subset \mathcal{N} \setminus Z$, and the submodularity and non-negativity of $f$ imply together that

$$f(B_\lambda \cup A) \ge f(A) + f(A \cup B_\lambda \cup Z) - f(A \cup Z) \ge f(A) - f(A \cup Z) \;. \qquad \square$$

With the above result, we are now ready to bound the expected value of $f(S_i)$.

**Lemma A.5.** *Let $\alpha = 1 - \frac{1}{k}$. Then, for every integer $0 \le i \le \lceil t_s \cdot k \rceil$,*

$$\mathbb{E}[f(S_i)] \ge (1 - \alpha^i) f(\mathbb{OPT} \setminus Z) - \left(1 - \alpha^i - i(1 - \alpha)\alpha^{i-1}\right) f(\mathbb{OPT} \cup Z) \;,$$

*and for every integer $\lceil t_s \cdot k \rceil \le i \le k$,*

$$\mathbb{E}[f(S_i)] \ge \frac{i - \lceil t_s \cdot k \rceil}{k} \alpha^{i - \lceil t_s \cdot k \rceil - 1} f(\mathbb{OPT}) - \frac{i - \lceil t_s \cdot k \rceil}{k} \left(\alpha^{i - \lceil t_s \cdot k \rceil - 1} - \alpha^{i-1}\right) f(\mathbb{OPT} \cup Z)$$
$$+ \alpha^{i - \lceil t_s \cdot k \rceil} \cdot f(S_{\lceil t_s \cdot k \rceil}) \;.$$

*Proof.* Let $E_i$ be an event fixing all the random decisions in Algorithm 5 up to iteration $i - 1$ (including), and let $A_i = \mathbb{OPT} \setminus Z$ for $i \le \lceil t_s \cdot k \rceil$ and $A_i = \mathbb{OPT}$ for $i > \lceil t_s \cdot k \rceil$. Since all the elements of $A_i$ can be chosen to be in $M_i$, and so can the dummy elements, we get that, conditioned on $E_i$

$$\mathbb{E}[f(u_i \mid S_{i-1})] \ge \max_{u \in M_i} f(u \mid S_{i-1}) \ge k^{-1} \sum_{u \in M_i} f(u \mid S_{i-1})$$

$$\ge k^{-1} \sum_{u \in A_i} f(u \mid S_{i-1}) \ge k^{-1} [f(S_{i-1} \cup A_i) - f(S_{i-1})] \;,$$

where the last inequality holds by the submodularity of $f$. Since $S_i = S_{i-1} + u_i$, rearranging the last inequality gives $\mathbb{E}[f(S_i)] \geq k^{-1}f(S_{i-1} \cup A_i) + \alpha f(S_{i-1})$. Taking expectation now over all possible events $E_i$ we get that, without conditioning on anything,

$$\mathbb{E}[f(S_i)] \geq k^{-1}\mathbb{E}[f(S_{i-1} \cup A_i)] + \alpha\mathbb{E}[f(S_{i-1})] \ . \tag{5}$$

**Proving the first inequality of the lemma.** We prove the first inequality of the lemma (for $0 \leq i \leq \lceil t_s \cdot k\rceil$) by induction on $i$. For $i = 0$ the inequality holds by the non-negativity of $f$. Assume now that $1 \leq i \leq \lceil t_s \cdot k\rceil$ and the inequality holds for $i-1$, and let us prove the inequality for $i$.

$$\mathbb{E}[f(S_j)] \geq k^{-1}\mathbb{E}[f(S_{i-1} \cup A_i)] + \alpha\mathbb{E}[f(S_{i-1})]$$

$$\geq k^{-1}\Big[f(\mathbb{OPT} \setminus Z) - \Big(1 - \Big(1 - \frac{1}{k}\Big)^{i-1}\Big) \cdot f(\mathbb{OPT} \cup Z)\Big]$$

$$+ \alpha(1 - \alpha^{i-1})f(\mathbb{OPT} \setminus Z) - \alpha\Big(1 - \alpha^{i-1} - (1-\alpha)(i-1)\alpha^{i-2}\Big)f(\mathbb{OPT} \cup Z)$$

$$= (1 - \alpha)\Big[f(\mathbb{OPT} \setminus Z) - (1 - \alpha^{i-1}) \cdot f(\mathbb{OPT} \cup Z)\Big]$$

$$+ \alpha(1 - \alpha^{i-1})f(\mathbb{OPT} \setminus Z) - \alpha\Big(1 - \alpha^{i-1} - (1-\alpha)(i-1)\alpha^{i-2}\Big)f(\mathbb{OPT} \cup Z)$$

$$= \alpha(1 - \alpha^i)f(\mathbb{OPT} \setminus Z) - \Big(1 - \alpha^i - (1-\alpha)i\alpha^{i-1}\Big)f(\mathbb{OPT} \cup Z) \ ,$$

where the second inequality holds by Lemma A.5 and the induction hypothesis since $A_i = \mathbb{OPT} \setminus Z$, and the first equality holds by definition of $\alpha$.

**Proving the second inequality of the lemma.** We prove the second inequality of the lemma (for $\lceil t_s \cdot k\rceil \leq i \leq k$) by induction on $i$. One can verify that for $i = \lceil t_s \cdot k\rceil$ the inequality trivially holds. Assume now that $\lceil t_s \cdot k\rceil < i \leq k$ and the inequality holds for $i-1$, and let us prove the inequality for $i$.

$$\mathbb{E}[f(S_j)] \geq k^{-1}\mathbb{E}[f(S_{i-1} \cup A_i)] + \alpha\mathbb{E}[f(S_{i-1})]$$

$$\geq k^{-1}\Big[\Big(1 - \frac{1}{k}\Big)^{i-\lceil t_s \cdot k\rceil-1} \cdot f(\mathbb{OPT}) - \Big(\Big(1 - \frac{1}{k}\Big)^{i-\lceil t_s \cdot k\rceil-1}$$

$$- \Big(1 - \frac{1}{k}\Big)^{i-1}\Big) \cdot f(\mathbb{OPT} \cup Z)\Big] + \frac{i - \lceil t_s \cdot k\rceil - 1}{k}\alpha^{i-\lceil t_s \cdot k\rceil-1}f(\mathbb{OPT})$$

$$- \frac{i - \lceil t_s \cdot k\rceil - 1}{k}\Big(\alpha^{i-\lceil t_s \cdot k\rceil-1} - \alpha^{i-1}\Big)f(\mathbb{OPT} \cup Z) + \alpha^{i-\lceil t_s \cdot k\rceil} \cdot f(S_{\lceil t_s \cdot k\rceil})$$

$$= k^{-1}[\alpha^{i-\lceil t_s \cdot k\rceil-1} \cdot f(\mathbb{OPT}) - (\alpha^{i-\lceil t_s \cdot k\rceil-1} - \alpha^{i-1}) \cdot f(\mathbb{OPT} \cup Z)]$$

$$+ \frac{i - \lceil t_s \cdot k\rceil - 1}{k}\alpha^{i-\lceil t_s \cdot k\rceil-1}f(\mathbb{OPT})$$

$$- \frac{i - \lceil t_s \cdot k\rceil - 1}{k}\Big(\alpha^{i-\lceil t_s \cdot k\rceil-1} - \alpha^{i-1}\Big)f(\mathbb{OPT} \cup Z) + \alpha^{i-\lceil t_s \cdot k\rceil} \cdot f(S_{\lceil t_s \cdot k\rceil})$$

$$= \frac{i - \lceil t_s \cdot k\rceil}{k}\alpha^{i-\lceil t_s \cdot k\rceil-1}f(\mathbb{OPT}) - \frac{i - \lceil t_s \cdot k\rceil}{k}\Big(\alpha^{i-\lceil t_s \cdot k\rceil-1} - \alpha^{i-1}\Big)f(\mathbb{OPT} \cup Z)$$

$$+ \alpha^{i-\lceil t_s \cdot k\rceil} \cdot f(S_{\lceil t_s \cdot k\rceil}) \ ,$$

where the second inequality holds by Lemma A.5 and the induction hypothesis since $A_i = \mathbb{OPT}$, and the first equality holds by definition of $\alpha$. $\qquad\square$

We are now ready to prove Theorem A.2.

*Proof of Theorem A.2.* Note that by submodularity and non-negativity of $f$,

$$f(\mathbb{OPT} \setminus Z) \geq f(\emptyset) + f(\mathbb{OPT}) - f(\mathbb{OPT} \cap Z) \geq f(\mathbb{OPT}) - f(\mathbb{OPT} \cap Z) \ .$$

The lower bound in the theorem follows by combining the above inequality with the following two inequalities: the inequality arising by plugging $i = \lceil t_s \cdot k\rceil$ into the first case of Lemma A.5, and the inequality arising by plugging $i = k$ into the second case of Lemma A.5.

To conclude the proof, observe that each one of the $k$ iterations of Algorithm 5 requires us to compute the marginal gain of at most $n$ elements with respect to the set $S_{i-1}$, which can be done using $O(n)$ queries to the objective function per iteration, and $O(nk)$ queries in total. $\qquad\square$

### A.3    0.385-Approximation Guarantee

We are now ready to present Algorithm 6 (the simpler version of our main algorithm). Recall that this algorithm returns the better among the two sets produced in the last two steps described in the beginning of this appendix. Formally, these sets are the output sets of LOCAL-SEARCH and Algorithm 5.

---

**Algorithm 6:** Warmup Algorithm

---

**input** : A positive integer $k \geq 1$, a non-negative submodular function $f$, and a flip point $0 \leq t_s \leq 1$
**output** : A set $S \subseteq \mathcal{N}$
1 $S_1 \leftarrow$ LOCAL-SEARCH$(k, f)$.
2 $S_2 \leftarrow$ the output of Algorithm 5.
3 **return** $\max\{f(Z), f(A)\}$.

---

The following theorem is proved by setting $\varepsilon$ in Algorithm 6 to be a small enough positive constant.

**Theorem A.6** (Approximation guarantee)**.** *Given an integer $k \geq 1$ and a non-negative submodular function $f\colon 2^{\mathcal{N}} \to \mathbb{R}_{\geq 0}$, there exists a 0.385-approximation algorithm for the problem of finding a set $S \subseteq \mathcal{N}$ of size at most $k$ maximizing $f$. This algorithm uses $O(nk^2)$ queries to the objective function.*

*Proof.* The proof of the approximation guarantee is very similar to the corresponding part in the proof Theorem 2.3, and is thus, omitted. The query complexity stated in the theorem follows directly from Theorems A.1 and A.2 and the fact that $\varepsilon$ is set to a positive constant value. $\qquad\square$

## B    Omitted Proofs of Section 2

In this section, we prove the theorems whose proofs have been omitted from Section 2, namely, Theorems 2.1, and 2.2.

### B.1    Proof of Theorem 2.1

In this section, we prove Theorem 2.1. We begin with the following lemma. We assume without loss of generality that $\mathbb{OPT}$ is of size $k$ (otherwise, we add to $\mathbb{OPT}$ dummy elements).

**Lemma B.1.** *If the set $S_0$ provides $c$-approximation, then each iteration of the loop starting on Line 3 in Algorithm 1 returns a set with probability at least $k/(c\varepsilon(1 - 1/e)L)$. Moreover, when this happens, the output set $S$ returned obeys*

$$f(S) \geq \frac{f(S \cap \mathbb{OPT}) + f(S \cup \mathbb{OPT})}{2 + \varepsilon} \quad and \quad f(S) \geq \frac{f(S \cap \mathbb{OPT})}{1 + \varepsilon} \ .$$

*Proof.* For every two integers $0 \leq i < L$ and $1 \leq j \leq \lceil \log \frac{1}{\varepsilon} \rceil$, we denote by $A_i^j$ the event that the set $S_i^j$ obeys the condition on Line 13 of Algorithm 1, i.e., the event that for every integer $0 \leq t \leq k$ it holds that

$$\max_{S \subseteq \mathcal{N} \setminus S_i^j, |S|=t} \sum_{u \in S} f(u \mid S_i^j) \leq \min_{S \subseteq S_i^j, |S|=t} \sum_{v \in S} f(v \mid S_i^j - v) + \varepsilon f(S_i^j) \ .$$

To better understand the implication of an event $A_i^j$, assume that such an event occurs, and observe that for $t = |\mathbb{OPT} \setminus S_i^j|$, it holds that

$$f(S_i^j) - f(S_i^j \cap \mathbb{OPT}) + \varepsilon f(S_i^j) = \sum_{\ell=1}^{|K_i^j|} f(u_\ell \mid J_i^j \cup \{u_1, \ldots, u_{\ell-1}\}) + \varepsilon f(S_i^j) \qquad (6)$$

$$\geq \sum_{\ell=1}^{|K_i^j|} f(u_\ell \mid J_L^j \cup K_L^j - u_\ell) + \varepsilon f(S_i^j)$$

$$= \sum_{v \in S_i^j \setminus \mathbb{OPT}} f(v \mid S_i^j - v) + \varepsilon f(S_i^j)$$

$$\geq \min_{S \subseteq S_i^j, |S|=t} \sum_{v \in S} f(v \mid S - v) + \varepsilon f(S_i^j)$$

$$\geq \max_{S \subseteq \mathcal{N} \setminus S_i^j, |S|=t} \sum_{u \in S} f(u \mid S_i^j)$$

$$\geq \sum_{u \in \mathbb{OPT} \setminus S_i^j} f(u \mid S_i^j) \geq f(S_i^j \cup \mathbb{OPT}) - f(S_i^j) \ ,$$

where the first equality holds by setting $J_i^j \triangleq S_i^j \cap \mathbb{OPT}$, $K_i^j \triangleq S_i^j \setminus \mathbb{OPT}$ and using $u_1, u_2, \ldots, u_{|K_i^j|}$ to denote the elements of $K_i^j$ in some arbitrary order. The first and last inequalities follow from submodularity of $f$, the second inequality holds since the fact that $S_i^j$ and $\mathbb{OPT}$ are both of size $k$ implies that $t = |\mathbb{OPT} \setminus S_i^j| = |S_i^j \setminus \mathbb{OPT}|$, and the third inequality holds under the assumption that event $A_i^j$ occurs. Rearranging the last inequality, we get

$$f(S_i^j) \geq \frac{f(S_i^j \cup \mathbb{OPT}) + f(S_i^j \cap \mathbb{OPT})}{2 + \varepsilon} \ .$$

In addition, the existence of the dummy elements implies that $\max_{S \subseteq \mathcal{N} \setminus S_i^j, |S|=t} \sum_{u \in S} f(u \mid S_i^j) \geq 0$, and plugging this inequality into Inequality (6) yields that the event $A_i^j$ also implies

$$f(S_i^j) - f(S_i^j \cap \mathbb{OPT}) + \varepsilon f(S_i^j) \geq \min_{S \subseteq S_i^j, |S|=t} \sum_{v \in S} f(v \mid S_i^j - v) + \varepsilon f(S_i^j) \geq 0 \ ,$$

and rearranging this inequality gives $f(S_i^j) \geq \frac{f(S_i^j \cap \mathbb{OPT})}{1 + \varepsilon}$.

The above shows that to prove the lemma it suffices to show that the probability that $A_{i^*}^j$ holds is at least $k/(c\varepsilon(1 - 1/e)L)$. Towards this goal, let us study the implications of the complementary event $\bar{A}_i^j$. Specifically, we would like to lower bound $\mathbb{E}\left[f(S_{i+1}^j) - f(S_i^j) \mid \bar{A}_i^j\right]$.

Fix a particular set $S_i^j$ that causes the event $\bar{A}_i^j$ to occur (notice that the occurrence of this event depends only on the set $S_i^j$). Then, there must exist sets $T_+ \subseteq \mathcal{N} \setminus S_i^j$ and $T_- \subseteq S_i^j$ of size $t \leq k$ such that

$$\sum_{u \in T_+} f(u \mid S_i^j) > \sum_{v \in T_-} f(v \mid S_i^j - v) + \varepsilon f(S_i^j) \ .$$

We can now define the event $B_i^j$ as the event that $Z_{i+1}^j \cap T_+ \neq \varnothing$ (notice that the event $B_i^j$ is defined only for this particular set $S_j^i$). The probability of the event $B_i^j$ is

$$\Pr(B_i^j \mid S_i^j) \geq 1 - \left(\frac{n - n/k}{n}\right)^{|T_+|} \geq 1 - e^{-\frac{|T_+|}{k}} \geq (1 - 1/e)\frac{|T_+|}{k} \ ,$$

where the first inequality holds since $(1 - \frac{1}{k})^x \leq e^{-\frac{x}{k}}$ for any $x \geq 0$, and the second inequality holds for any $x \in [0, 1]$ by the concavity of $1 - e^{-x}$. By the law of total expectation and the fact that $f(S_{i+1}^j)$ is always at least $f(S_i^j)$, we now get

$$\mathbb{E}\left[f(S_{i+1}^j) - f(S_i^j) \mid S_i^j\right] \geq \Pr(B_i^j \mid S_i^j) \cdot \mathbb{E}\left[f(S_{i+1}^j) - f(S_i^j) \mid S_i^j, B_i^j\right]$$

$$\geq (1 - 1/e)\frac{|T_+|}{k} \cdot \mathbb{E}\left[f(S_{i+1}^j) - f(S_i^j) \mid S_i^j, B_i^j\right]$$

$$\geq \frac{1 - 1/e}{k}\varepsilon c f(\mathbb{OPT}) \ ,$$

where the last inequality holds since

$$\mathbb{E}\Big[f(S_{i+1}^j) - f(S_i^j) \mid S_i^j, B_i^j\Big] \geq \mathbb{E}\Big[f(S_i^j - v_{i+1}^j + u_{i+1}^j) - f(S_i^j) \mid S_i^j, B_i^j\Big]$$

$$= \mathbb{E}\Big[f(S_i^j - v_{i+1}^j + u_{i+1}^j) - f(S_i^j - v_{i+1}^j) + f(S_i^j - v_{i+1}^j) - f(S_i^j) \mid S_i^j, B_i^j\Big]$$

$$\geq \mathbb{E}\Big[f(S_i^j + u_{i+1}^j) - f(S_i^j) + f(S_i^j - v_{i+1}^j) - f(S_i^j) \mid S_i^j, B_i^j\Big]$$

$$\geq \mathbb{E}\Big[f(S_i^j + u') - f(S_i^j) \mid S_i^j, B_i^j\Big] + \mathbb{E}\Big[f(S_i^j - v') - f(S_i^j) \mid S_i^j, B_i^j\Big]$$

$$= \frac{\sum\limits_{u \in T_+} f(u \mid S_i^j)}{|T_+|} - \frac{\sum\limits_{v \in T_-} f(v \mid S_i^j - v)}{|T_-|} \geq \frac{\varepsilon f(S_i^j)}{|T_+|} \geq \frac{c\varepsilon f(\mathbb{OPT})}{|T_+|} \ ,$$

where the second inequality holds by submodularity of $f(\cdot)$, the third inequality holds by the way Algorithm 1 chooses $u_i^j$ and $v_i^j$ if we let $u'$ be a uniformly random element of $T_+ \cap Z_{i+1}^j$ and $v'$ be a uniformly random element of $T_-$, the penultimate inequality holds by our assumption that $S_i^j$ implies the event $\bar{A}_i^j$, and finally, the last inequality holds since it is guaranteed that $f(S_i^j) \geq f(S_0^j) = f(S_0) \geq c \cdot f(\mathbb{OPT})$.

Since the above bound on the expectation holds conditioned on every set $S_i^j$ that implies the event $\bar{A}_i^j$, it holds (by the law of total expectation) also conditioned on the event $\bar{A}_i^j$ itself. Adding this lower bound for all $i$ values, and using the non-negativity of $f$, we get

$$\mathbb{E}\Big[f(S_L^j)\Big] \geq \sum_{\ell=0}^{L-1} \mathbb{E}\Big[f(S_{\ell+1}^j) - f(S_\ell^j)\Big] \geq \sum_{\ell=0}^{L-1} \Pr(\bar{A}_\ell^j) \cdot \mathbb{E}\Big[f(S_{\ell+1}^j) - f(S_\ell^j) \mid \bar{A}_\ell^j\Big]$$

$$\geq \frac{1 - 1/e}{k} c\varepsilon f(\mathbb{OPT}) \cdot \sum_{\ell=1}^{L} \Pr(\bar{A}_\ell^j) \ .$$

Combining the last inequality with the fact that $f(S_L^j)$ is deterministically at most $f(\mathbb{OPT})$, it must hold that

$$1 \geq \frac{1 - 1/e}{k} c\varepsilon \cdot \sum_{\ell=1}^{L} \Pr\Big(\bar{A}_\ell^j\Big) \quad \Rightarrow \quad \sum_{\ell=1}^{L} \Pr\Big(\bar{A}_\ell^j\Big) \leq \frac{k}{c\varepsilon(1 - 1/e)} \ .$$

Hence, the probability that the event $A_{i^*}^j$ does not hold for a uniformly random $i^* \in [L]$ is

$$\frac{\sum_{\ell=1}^{L} \Pr\Big(\bar{A}_\ell^j\Big)}{L} \leq \frac{k}{c\varepsilon(1 - 1/e)L} \ . \qquad \qquad \square$$

**Theorem 2.1.** *There exists an algorithm that given a positive integer $k$, a value $\varepsilon \in (0, 1)$, and a non-negative submodular function $f : 2^{\mathcal{N}} \to \mathbb{R}_{\geq 0}$, outputs a set $S \subseteq \mathcal{N}$ of size at max $k$ that, with probability at least $1 - \varepsilon$, obeys*

$$f(S) \geq \frac{f(S \cap \mathbb{OPT}) + f(S \cup \mathbb{OPT})}{2 + \varepsilon} \quad \text{and} \quad f(S) \geq \frac{f(S \cap \mathbb{OPT})}{1 + \varepsilon} \ .$$

*Furthermore, the query complexity of the above algorithm is $O_\varepsilon(n + k^2)$.*

*Proof.* As mentioned above, we initialize the set $S_0$ using the deterministic $1/4$-approximation algorithm of Balkanski et al. [2], which uses only $O(n)$ queries to the objective function. Thus, $c = 1/4$ in our implementation of Algorithm 1. Let us now set $L = \left\lceil \frac{2k}{c\varepsilon(1 - 1/e)} \right\rceil$ in Algorithm 1. Then, Lemma B.1 guarantees that every iteration of the outer loop of the algorithm returns a set (obeying the requirement of the theorem) with probability at least $1/2$. Hence, by repeating this loop $\lceil \log_2 \varepsilon^{-1} \rceil$ times, we are guaranteed that Algorithm 1 outputs a set with probability at least $1 - \varepsilon$. To complete the proof of the theorem, it only remains to bound the number of queries to the objective function that are necessary for implementing it. Each iteration of Algorithm 1 can be

implemented using $O(n/k + k)$ queries, and for the above choices of $L$, Algorithm 1 has only $O_\varepsilon(k)$ iterations. Thus, all the iterations of the algorithm can be implemented using $O_\varepsilon(n + k^2)$ queries in total. It should also be mentioned that evaluating the condition on Line 13 of the algorithm requires $O(n + k)$ queries to the objective, and since this condition is evaluated $\lceil \log_2 \varepsilon^{-1} \rceil = O_\varepsilon(1)$ times, all its evaluations require in total only $O_\varepsilon(n + k)$ queries. $\qquad\square$

### B.2 Proof of Theorem 2.2

In this section, we prove Theorem 2.2. We begin by observing that, like in Algorithm 5, each iteration of Algorithm 2 adds each element $u \in \mathcal{N}$ into the solution with probability at most $1/k$, and furthermore, the first $\lceil t_s \cdot k \rceil$ iterations of the algorithm do not pick elements of $Z$ at all (see the analysis of Sample Greedy in [6] for a proof of a similar observation that is given in more detail). Given this observation, the proof of Lemma A.4 applies also to Algorithm 2. Thus, this lemma can be used in the proof of the following result.

**Lemma B.2.** *Let $\alpha = 1 - \frac{1}{k}$. Then, for every integer $0 \le i \le \lceil t_s \cdot k \rceil$,*

$$\mathbb{E}[f(S_i)] \ge \left(1 - \alpha^i\right) f(\mathbb{OPT} \setminus Z) - \left(1 - \alpha^i - i(1 - \alpha)\alpha^{i-1}\right) f(\mathbb{OPT} \cup Z) - \frac{2\varepsilon i}{k} \quad,$$

*and for every integer $\lceil t_s \cdot k \rceil \le i \le k$,*

$$\mathbb{E}[f(S_i)] \ge \frac{i - \lceil t_s \cdot k \rceil}{k} \alpha^{i - \lceil t_s \cdot k \rceil - 1} f(\mathbb{OPT}) - \frac{i - \lceil t_s \cdot k \rceil}{k} \left(\alpha^{i - \lceil t_s \cdot k \rceil - 1} - \alpha^{i-1}\right) f(\mathbb{OPT} \cup Z)$$
$$+ \alpha^{i - \lceil t_s \cdot k \rceil} \cdot f(S_{\lceil t_s \cdot k \rceil}) - \frac{2\varepsilon(i - \lceil t_s \cdot k \rceil)}{k} \quad.$$

*Proof.* Let $E_i$ be an event fixing all the random decisions in Algorithm 2 up to iteration $i - 1$ (including), and let $A_i = \mathbb{OPT} \setminus Z$ for $i \le \lceil t_s \cdot k \rceil$ and $A_i = \mathbb{OPT}$ for $i > \lceil t_s \cdot k \rceil$. Since all the elements of $A_i$ can be sampled in iteration $i$, by following the proof of Lemma 13 in the analysis of the Sample Greedy algorithm by [6], one can obtain that, conditioned on $E_i$,

$$\mathbb{E}[\max\{0, f(u_i \mid S_{i-1})\}] \ge \frac{1 - \varepsilon}{k} [f(A_i \cup S_{i-1}) - f(S_{i-1})] \quad.$$

To be more specific, Lemma 11 of [6] shows that with probability at least $1 - \varepsilon$ the element chosen as $u_i$ in iteration $i$ of Algorithm 2 belongs to the $k$ elements with the largest marginal values among the elements that can be sampled in this iteration (if less than $k$ elements can be sampled, dummy elements should added for the purpose of this argument). Let $B_i$ denote the set of these $k$ elements. Since the probability of each element of $B_i$ to be selected as $u_i$ is non-decreasing in $f(u_i \mid S_{i-1})$, by Chebyshev's sum inequality, we get

$$\mathbb{E}[\max\{0, f(u_i \mid S_{i-1})\}] \ge (1 - \varepsilon) \frac{\sum_{u \in B_i} \max\{0, f(u \mid S_{i-1})\}}{k}$$
$$\ge (1 - \varepsilon) \frac{\sum_{u \in A_i} f(u \mid S_{i-1})}{k} \ge \frac{1 - \varepsilon}{k} [f(A_i \cup S_{i-1}) - f(S_{i-1})] \quad,$$

where the second inequality holds since $B_i$ contains the $k$ elements with the largest marginals among the elements that can be samples, and $A_i$ is a set of up to $k$ such elements; and the last inequality follows from the submodularity of $f$.

Since $S_i = S_{i-1} + u_i$ when $f(u_i \mid S_{i-1}) \ge 0$ and $S_i = S_{i-1}$ otherwise, we get $f(S_i) - f(S_{i-1}) = \max\{0, f(u_i \mid S_{i-1})\}$. Plugging this observation into the previous inequality, and rearranging gives

$$\mathbb{E}[f(S_i)] \ge \frac{1 - \varepsilon}{k} f(S_{i-1} \cup A_i) + \left(1 - \frac{1 - \varepsilon}{k}\right) f(S_{i-1})$$
$$\ge \frac{1}{k} f(S_{i-1} \cup A_i) + \alpha f(S_{i-1}) - \frac{\varepsilon}{k} f(S_{i-1} \cup A_i)$$
$$\ge \frac{1}{k} f(S_{i-1} \cup A_i) + \alpha f(S_{i-1}) - \frac{2\varepsilon}{k} f(\mathbb{OPT}) \quad,$$

where the second inequality uses the non-negativity of $f$, and the last inequality holds since $f(S_{i-1} \cup A_i) \le f(S_{i-1}) + f(A_i) - f(S_{i-1} \cap A_i) \le f(S_{i-1}) + f(A_i) \le 2f(\mathbb{OPT})$ because both $S_{i-1}$ and

$A_i$ are feasible solutions, and thus, cannot have a value larger than $f(\mathbb{OPT})$. Taking expectation now over all possible events $E_i$ we get that, without conditioning on anything,

$$\mathbb{E}[f(S_i)] \geq k^{-1}\mathbb{E}[f(S_{i-1} \cup A_i)] + \alpha\mathbb{E}[f(S_{i-1})] - \frac{2\varepsilon}{k}f(\mathbb{OPT}) \ .$$

The remaining part of this proof is omitted since it is very similar to the corresponding part in the proof of Lemma A.5, except that the last inequality should be used instead of Inequality (5).   □

We are now ready to prove Theorem 2.2.

**Theorem 2.2.** *There exists an algorithm that given a positive integer $k$, a value $\varepsilon \in (0, 1)$, a value $t_s \in [0, 1]$, a non-negative submodular function $f \colon 2^{\mathcal{N}} \to \mathbb{R}_{\geq 0}$, and a set $Z \subseteq \mathcal{N}$ obeying the inequalities stated in Theorem 2.1, outputs a solution $S_k$, obeying*

$$\mathbb{E}[f(S_k)] \geq \Big( \frac{k - \lceil t_s \cdot k \rceil}{k}\alpha^{k - \lceil t_s \cdot k \rceil - 1} + \alpha^{k - \lceil t_s \cdot k \rceil} - \alpha^k \Big)f(\mathbb{OPT}) +$$
$$+ \Big( \alpha^k + \alpha^{k-1} - \frac{2k - \lceil t_s \cdot k \rceil}{k}\alpha^{k - \lceil t_s \cdot k \rceil - 1} \Big)f(\mathbb{OPT} \cup Z)$$
$$+ (\alpha^k - \alpha^{k - \lceil t_s \cdot k \rceil})f(\mathbb{OPT} \cap Z) - 2\varepsilon f(\mathbb{OPT}) \ ,$$

*where $\alpha \triangleq 1 - 1/k$. Moreover, this algorithm requires only $O_\varepsilon(n)$ queries to the objective function.*

*Proof of Theorem 2.2.* The lower bound in Theorem 2.2 follows from the same arguments used in the proof of Theorem A.2, except that Lemma B.2 is used instead of Lemma A.5, which results in the additional error term $2\varepsilon f(\mathbb{OPT})$ in the lower bound of Theorem 2.2.

To bound the number of queries to the objective function necessary for implementing Algorithm 2, observe that each iteration of Algorithm 2 samples $O_\varepsilon(n/k)$ elements, and the marginal gain (with respect to $S_{i-1}$) has to be computed only for the sampled elements. Thus, each iteration of Algorithm 2 requires only $O_\varepsilon(n/k)$ queries to the objective function. Since the algorithm has only $k$ iterations, its total query complexity is $k \cdot O_\varepsilon(n/k) = O_\varepsilon(n)$.   □

