# OpenReview forum: "Practical $0.385$-Approximation for Submodular Maximization Subject to a Cardinality Constraint"
_NeurIPS.cc/2024/Conference — NeurIPS 2024 poster_

### Official Review · Reviewer_2jpf · 2024-06-25

**Soundness:** 3
**Presentation:** 2
**Contribution:** 3
**Rating:** 6
**Confidence:** 2

**Summary:**

The paper studies the problem of approximately maximizing a submodular function under a constraint that the sets can be of size at most $k$. By carefully combining algorithms from previous work and developing them further, the authors obtain an algorithm that guarantees a $0.385$-approximation with $O_\epsilon(n + k^2)$ queries to the function and a user-defined probability of failure. The good performance of the proposed algorithm is demostrated with several experiments.

**Strengths:**

Even though the work relies heavily on previous work as described in the beginning of Sec. 3, their combination appears highly nontrivial to me, and the techniques have been developed further from the previous work.

The empirical results demonstrate clear improvements over previous state of the art and appear to have significantly less variance in the output.

**Weaknesses:**

Because of my limited expertise on the topic, my largest complaint is on the presentation of the results: Although the authors do a good job in explaining the ideas behind the math, the formulas are occasionally relatively cumbersome to read, e.g., Line 14 of Algorithm 1. On the other hand, I must admit that I do not have an immediate solution on how to improve them.

**Questions:**

In terms of the running time, i.e., not the number of queries, how does the proposed algorithm compare against the state of the art? Because the running time depends on, for example, $epsilon$, it is not immediately clear to me how large the constant factors of the algorithms would be.

Other:
- Abstract: Without previous background on the topic, it was unclear to what $k$ and $n$ referred to in the abstract.
- Lines 29–32: first, $B$ is a subset of $A$, but then $A$ is a subset of $B$. This can be confusing to the reader.
- Line 66: If I understood the paper correctly, I think it would be more precise to use $O_\epsilon$ instead of the $O$-notation here (although a constant $\epsilon$ is used in the experiments)
- Caption of Alg. 3: maximiziation -> maximization
- Line 172: OPT is not written with \mathbb

**Limitations:**

Since the paper focuses on a optimizing a method that is used in ML applications and does not study application directly, I think the limited discussion of limitations is sufficient here.

---

> ### Author Rebuttal · Authors · 2024-08-06
>
> We thank the reviewer for the positive evaluation of our work. We have thoroughly addressed each of the reviewer’s raised concerns and are eager to engage in further discussion to ensure all remaining issues are resolved.
>
>   1. **In terms of the running time, i.e., not the number of queries, how does the proposed algorithm compare against the state of the art? Because the running time depends on, for example, $epsilon$, it is not immediately clear to me how large the constant factors of the algorithms would be.**
>
> The runtime is a problematic measure as it highly depends on implementation details and the level of optimization employed, often making comparisons based on this method non-robust. Thus, it is customary in the literature about submodular maximization to use the number of queries to the objective function as a more reliable proxy for the real-world performance of an algorithm. Accordingly, we include in the PDF of the general rebuttal a plot comparing the number of queries used by our algorithm and central algorithms from the literature.
> Interestingly, the number of queries used by our algorithm is empirically dominated by the queries used for the initialization of the Fast Local search procedure (Line 1 of Algorithm 1). Implementing this initialization in the way described in the submitted paper leads to a query complexity that is somewhat worse than that of other algorithms. However, an alternative implementation of this initialization is implied by a very recent paper [1] that describes a fast ¼-approximation algorithm for maximizing non-monotone submodular functions under matroid constraints (of which cardinality constraints are a special case). When initialized this way, our algorithm becomes much faster (in terms of the empirical number of query calls), while the quality of the solution produced in terms of the objective value remains almost unchanged. We stress that the change in the initialization method only affects the implementation of Line 1 of Algorithm 1 in our paper.
>
>
> 	[1] Balkanski, Eric, Steven DiSilvio, and Alan Kuhnle. "Submodular Maximization in Exactly n Queries" arXiv preprint arXiv:2406.00148 (2024).
>
>
>   2. **Typos/Writing suggestions/comments**
>
> We thank the reviewer for their keen observations. We have addressed your comments and revised the manuscript accordingly.

---

> > ### Comment · Reviewer_2jpf · 2024-08-08
> >
> > Although I partially agree with your criticism of evaluating the running time, I would also argue that it is still one factor to keep in mind. Consider, e.g., fast matrix multiplication, where the asymptotical speedups are mostly of theoretical interest due to the large constant factors hidden away by the O-notation.
> >
> > I'm sufficiently happy with the Rebuttal and will keep my score.

---

### Official Review · Reviewer_yvAe · 2024-07-05

**Soundness:** 2
**Presentation:** 2
**Contribution:** 2
**Rating:** 5
**Confidence:** 4

**Summary:**

The paper introduces a new algorithm, FAST-LOCAL-SEARCH, for maximizing non-monotone submodular functions, achieving a 0.385-approximation with low query complexity. It combines initial solution search, accelerated local search, and stochastic greedy improvement steps to outperform existing algorithms in machine-learning applications like movie recommendation, image summarization, and revenue maximization. The algorithm guarantees a constant approximation to the optimal set and is supported by theoretical proofs and empirical evaluations.

**Strengths:**

The paper introduces an algorithm, FAST-LOCAL-SEARCH, for maximizing non-monotone submodular functions under a cardinality constraint. This algorithm combines initial solution search, accelerated local search, and stochastic greedy improvement steps to achieve a 0.385-approximation with query complexity of O(n + k^2). The practical performance of the algorithm has been validated in real-world applications.

**Weaknesses:**

The novelty of the paper is limited. The idea of this paper is derived from existing work [2], and more significantly, the "guided algorithm" proposed in this paper is very similar to [10] in both algorithm design and theoretical analysis.

The authors claim that the algorithm proposed in this paper has a practical query complexity of O(n+k^2); however, in practical applications, the value of k is often very large, making an O(k^2) query complexity potentially impractical.

**Questions:**

Please refer to weaknesses.

**Limitations:**

N.A.

---

> ### Author Rebuttal · Authors · 2024-08-06
>
> We would like to thank the reviewer for the detailed review. We hope that our response below addresses all their concerns about the paper. If further clarification is needed, we will be happy to provide it.
>
>   1. **The novelty of the paper is limited. The idea of this paper is derived from existing work [2], and more significantly, the "guided algorithm" proposed in this paper is very similar to [10] in both algorithm design and theoretical analysis.**
>
> As mentioned by the reviewer, our algorithm is based on ideas from [2], and its main novelty is in finding a way to implement these ideas efficiently and practically. As explained in Section 1.1, the work of [10] is an independent parallel work that is also based on [2]. Both works come up with a very similar basic algorithm having a query complexity of O(nk). However, the two works diverge beyond this point. The objective of [10] was to derandomize this basic algorithm and extend it to other constraints. In contrast, our objective was to further speed up the algorithm beyond the above-mentioned query complexity. This additional speed-up is an important technical contribution of our work. For example, getting the local search part of the algorithm of [2] to run fast (as implemented in Algorithm 1) required us to relax the notion of local optimum used and introduce a sophisticated probabilistic analysis deviating from the well-threaded path of analysis of local search algorithms in submodular optimization.
>
>
>   2. **The authors claim that the algorithm proposed in this paper has a practical query complexity of O(n+k^2); however, in practical applications, the value of k is often very large, making an O(k^2) query complexity potentially impractical.**
>
>    As mentioned by the reviewer, the query complexity of our algorithm is O(n + k^2), which is linear for k = O(n^{½}). For applications that require larger values of k, our algorithm indeed runs in super-linear time, which is unfortunate. Still, one should note two things:
>    - The PDF of the general rebuttal includes a plot comparing the empirical query complexity of our algorithm with various existing algorithms, including linear query complexity algorithms like the one of [5]. This plot demonstrates that our algorithm compares very well with these algorithms in terms of the empirical query complexity (while outcompeting them in terms of the value of the solution produced).
>
>    - The only other algorithm in the literature that guarantees an approximation ratio better than 1/e and has a possibly practical query complexity is the algorithm due to the recent independent work of [10]. This algorithm has a query complexity of $O(nk)$, which is worse than the query complexity of our algorithm except in the regime of k = $\Omega(n)$ (and in this regime the query complexities of both algorithms become identical).

---

> ### Comment · Reviewer_yvAe · 2024-08-12
>
> Thank you for your response. I decide to increase my score.

---

### Official Review · Reviewer_q8Tb · 2024-07-11

**Soundness:** 2
**Presentation:** 4
**Contribution:** 2
**Rating:** 6
**Confidence:** 4

**Summary:**

This work addresses the problem of maximizing a non-monotone submodular function subject to a cardinality constraint. In submodular maximization, we have a set of elements (ground set) and a function that assigns a value to any subset of elements. A function is submodular if adding an element to a smaller set contributes more value than adding it to a larger set. Formally, for sets A and B where A is a subset of B, adding an element x to A increases the function value more than adding x to B. A function is monotone if its value always increases with set size. Non-monotone functions don't necessarily have this property. The goal here is to find a subset of size at most k from the ground set that maximizes the value of the non-monotone submodular function.

While they developed a 0.385-approximate solution for this problem, there is an algorithm with a better approximation factor of 0.401. Another work also achieves a 0.385-approximate solution, but both are impractical due to the high number of query calls required. In contrast, this work uses only O(n+k^2) query calls. Additionally, there are two works with a 0.367-approximation, one requiring O(nk) query calls and the other O(n) query calls. In the experiment section, the authors compared the output value of their algorithm with those of the two latter algorithms.

The core idea of the proposed algorithm involves:
- Running the sample greedy algorithm by Buchbinder et al. [5].
- Applying a local search technique to improve the initial solution.
- Employing a Stochastic greedy method and returning the solution with the highest value between the local search and Stochastic greedy outputs.

This approach leverages existing techniques while incorporating additional steps to enhance the solution within a reasonable computational cost.

**Strengths:**

For non-monotone submodular maximization under a cardinality constraint, their work achieves the best query complexity for algorithms that attain at least a 0.385-approximate solution. In other words, among practical algorithms—those that can be implemented and executed in a reasonable time—they offer the highest approximation factor.

**Weaknesses:**

Since their algorithm does not have the best approximation factor, it is less interesting for theoretical work and more important for practical applications. Therefore, a comprehensive experiment supporting their claims is crucial. While they did a great job by running experiments on different problems and demonstrating the advantage of their output value compared to two other algorithms with worse approximation guarantees, it is also necessary to plot the number of query calls for these algorithms. Comparing the number of query calls in addition to the output value is essential, as they claimed to have the best approximation algorithm with "a low and practical" query complexity.

### Comments for the authors:
- Line 43: Remove “over”
- Line 50: I think [14] present a ½-approximation algorithm for symmetric submodular functions and for non-symmetric, they only achieve a ⅖-approximation algorithm.
- Line 68: Do you mean “3 applications”?
- Line 149: Remove “)” after “pn”
- Line 208: Replace “she is” with “they are”
- Line 548: Replace “lemmata” with “lemmas”
The revenue maximization experiment is mostly known as the max-cut problem. I think it is good to at least mention that.
- In Algorithm 3, the term “AIDED-MEASURED DISCRETE STOCHASTIC GREEDY” has been used to refer to Algorithm 2. However, the name of Algorithm 2 is different.
- From lines 95 and 96, I assumed that Algorithm 2 improves the output of Algorithm 1. However, after reading the algorithms, it seems that Algorithm 2 finds a different

**Questions:**

- What is the query complexity of your algorithm in terms of epsilon?
- Do you have any practical comparisons of the number of query calls used in your experiments?

---

> ### Author Rebuttal · Authors · 2024-08-06
>
> We are grateful for the reviewer’s positive evaluation of our work. In what follows, we address the concerns raised by the reviewer in detail. We will be happy to engage to address any lingering concerns.
>
>   1. **Comparing the number of query calls in addition to the output value is essential, as they claimed to have the best approximation algorithm with "a low and practical" query complexity. | Do you have any practical comparisons of the number of query calls used in your experiments?**
>
> We thank the reviewer for the insightful suggestion. We have generated graphs depicting the number of query calls, comparing the efficiency of our method with that of our competitors. These graphs can be found in the PDF file of the general rebuttal.
> Interestingly, the number of queries used by our algorithm is empirically dominated by the queries used for the initialization of the Fast Local search procedure (Line 1 of Algorithm 1). Implementing this initialization in the way described in the submitted paper leads to a query complexity that is somewhat worse than that of other algorithms. However, an alternative implementation of this initialization is implied by a very recent paper [1] that describes a fast ¼-approximation algorithm for maximizing non-monotone submodular functions under matroid constraints (of which cardinality constraints are a special case). When initialized this way, our algorithm becomes much faster (in terms of the empirical number of query calls), while the quality of the solution produced in terms of the objective value remains almost unchanged. We stress that the change in the initialization method only affects the implementation of Line 1 of Algorithm 1 in our paper.
>
>
> 	[1] Balkanski, Eric, Steven DiSilvio, and Alan Kuhnle. "Submodular Maximization in Exactly n Queries" arXiv preprint arXiv:2406.00148 (2024).
>
>
>   2. **What is the query complexity of your algorithm in terms of epsilon?**
>
> The query complexity of our algorithm is $O\left(n * \epsilon^{-2} \log\left(\frac{1}{\varepsilon}\right) + k^2 * \varepsilon^{-1} \log\frac{1}{\varepsilon}\right)$, excluding the query complexity required for obtaining the initial solution $S_0$ for the Fast Local search procedure. The query complexity required for getting this initial solution is independent of epsilon when the algorithm of [1] is used for that purpose, as described in the response to the previous question.
>
>
>   3. **From lines 95 and 96, I assumed that Algorithm 2 improves the output of Algorithm 1. However, after reading the algorithms, it seems that Algorithm 2 finds a different**
>
>
> Intuitively, Algorithms 1 and 2 balance each other. In a sense, Algorithm 2 uses the output of Algorithm 1 as a “set to avoid”, and its analysis shows that when the output of Algorithm 1 has a poor value, then by avoiding it Algorithm 2 is guaranteed to output a set of good value. Thus, it is guaranteed that the better of the outputs of the two algorithms is always a good solution.
>
>   4. **Typos/writing suggestions/comments**
>
> We have addressed all of the raised comments and suggestions concerning the writing. We are thankful for the keen observation of the reviewer and the fruitful suggestions.

---

> > ### Comment · Reviewer_q8Tb · 2024-08-10
> >
> > Thank you for your response. Since my questions have been fully answered and the plots comparing oracle calls have been provided, I’ve increased my score to 6.

---

### Official Review · Reviewer_USjH · 2024-07-13

**Soundness:** 4
**Presentation:** 3
**Contribution:** 3
**Rating:** 7
**Confidence:** 5

**Summary:**

The paper presents a practical $0.385$-approximation algorithm using $O(n+k^2)$ queries for non-monotone submodular maximization under a cardinality (size) constraint, where $n$ is the number of elements in the ground set and $k$ is the maximum size of a feasible solution. As a comparison, the state-of-the-art algorithms attain a $0.401$ approximation ratio with a high number of queries, or $1/e-\epsilon$ approximation ratio using $O_{\epsilon}(n)$ queries. The paper also evaluates its method experimentally on several applications.

**Strengths:**

1.	The paper makes a substantial progress for the studied problem by presenting a new algorithm with low query complexity and improved approximation ratio. The algorithm is obtained by carefully combining several well-known techniques and methods in the field. Such a result already contains enough originality as a NeurIPS submission.
2.	The paper is complete, in the sense that it contains both a correct theoretical analysis and detailed experimental evaluations of its algorithm.
3.	The paper is organized well. It splits the algorithm into several ingredients and then explains each part in detail and how to combine them to get the final result. In such a way, it’s easy for readers to follow the basic ideas and check the correctness of the algorithm.
4.	The paper, as suggested by the experiments, provides a better solution for several important applications. On the other hand, it further exploits several mature techniques in the field, which may inspire more results in the area.

**Weaknesses:**

1.	There are many typos in the proofs (see Questions for some examples), which affects the clarity. The authors are suggested to proofread their paper in a later version.
2.	The main algorithmic framework is borrowed from [2]. So, if the space is enough, I suggest the authors to add a paragraph discussing why the framework can beat the $1/e$ ratio. In doing so, the paper might be more readable to those who haven’t read [2] before.

**Questions:**

1.	I’m curious about if it’s possible to discretize the algorithm in [3] to get a practical $0.401$ algorithm, just like this paper.
2.	Following are some typos:
1) I suggest using “at most” to replace “at max” in the paper.
2) Page 13. “Let S be a subset in N of size at (missing most) k”.
3) Page 16, Lemma A.3, $x_{\in}$ --> $x\in$.
4) Page 16, in the last inequality, \beta^{i} should be \beta_{i}.
5) Page 17, in the first formula, A\cup B_{\lambda}\cap Z should be A\cup B_{\lambda} ‘\cup’ Z.
6) Page 18, Lemma B.1, “that starts returns a set…” doesn’t read smoothly.
7) Page 18, in the first sentence of the proof of Lemma B.1, S_{i,j} should be S_i^j.
8) Page 20, in inequality (6), the subscript L should be i.

**Limitations:**

Yes

---

> ### Author Rebuttal · Authors · 2024-08-06
>
> We deeply appreciate the reviewer's meticulous evaluation of our paper. Thank you for your invaluable feedback. In what follows, we address the reviewer’s questions/comments.  We will be happy to engage with you to address any lingering concerns
>
>   1. **There are many typos in the proofs (see Questions for some examples), which affects the clarity. The authors are suggested to proofread their paper in a later version.**
>
>
> We have addressed the typos raised in the Questions section and will thoroughly pass over the paper to fix any additional types. We thank you for your keen observation.
>
>   2. **The main algorithmic framework is borrowed from [2]. So, if the space is enough, I suggest the authors to add a paragraph discussing why the framework can beat the $\frac{1}{e}$ ratio. In doing so, the paper might be more readable to those who haven’t read [2] before.**
>
>
> We thank the reviewer for the fruitful suggestion. We will be more than happy to add such a paragraph to the paper.
>
>
>   3. **I’m curious about if it’s possible to discretize the algorithm in [3] to get a practical $0.401$ algorithm, just like this paper.**
>
> Making the recent 0.401-approximation of [3] practical is an interesting question that we have also considered. Unfortunately, this goal seems to be very challenging and requires significant new ideas because the algorithm of [3] has a query complexity that is exponential in 1/epsilon. Notice that this complexity remains exponential even if we assume access to the multilinear extension of the objective function rather than requiring the algorithm to use samples to estimate it. Making the algorithm combinatorial, as we have done in the current paper, can make this assumption justified, but it does not help to alleviate the inherent exponentiality of the query complexity of the other parts of the algorithm.

---

> > ### Comment · Reviewer_USjH · 2024-08-13
> >
> > Thank you for your response. I will keep my score.

---

### Author Rebuttal · Authors · 2024-08-06

We sincerely appreciate the reviewers for their constructive feedback and insightful questions about our research. Your dedication and expertise in helping us enhance our work is invaluable. We are grateful for your observations and thank you for your contributions.
Finally, attached here is our PDF containing all the figures depicting our algorithm with a different initialization in comparison with our algorithm with the original initialization discussed in the appendix and also in comparison with the presented competitors in the paper. We present in these figures a comparison with respect to two metrics:
  - Objective value, and
  - Oracle calls


The initialization we are referring to is from [1] and is referred to as "Algorithm 1 with different initialization" in the uploaded graphs.

    [1] Balkanski, Eric, Steven DiSilvio, and Alan Kuhnle. "Submodular Maximization in Exactly n Queries" arXiv preprint arXiv:2406.00148 (2024).

---

### Comment · Area_Chair_1Md8 · 2024-08-12

Reviewers: The author-reviewer discussions end tomorrow (August 13), so please read the authors' rebuttal carefully and respond if you haven't.

---

### Author Response · Authors · 2024-08-14
**Thank you!**

We sincerely thank the reviewers for their careful reading, insightful comments, and active engagement during the authors-reviewers rebuttal period. Your feedback has been invaluable in enhancing our paper, and we greatly appreciate your input.

Best regards,
The Authors.

---

### Decision · Program_Chairs · 2024-09-25

**Decision:**

Accept (poster)

**Comment:**

The ratings for this paper are {7, 6, 5, 6}. Moreover, Reviewer USjH is willing to champion the paper (score 7).

**Remark 1.** I checked DBLP and Reviewer USjH has not published a paper with any authors of this submission (arXiv:2405.13994).

**Decision.** I think we should accept this paper on the grounds that all reviewers lean accept.

**Remark 2.** That said, I read the paper myself prior to the reviews and was hoping it would be simpler and cleaner than it ended up being (since it doesn't advance the SotA approximation ratio). My personal rating would be 4 (lean reject).